



# Uncertainty and retrieval sensitivity in TROPOMI-based methane inversions over the North Slope of Alaska

Rebecca H. Ward[1,2], Luke M. Western[3], Rachel L. Tunnicliffe[3], Elena Fillola[4], Aki Tsuruta[2], Tuula Aalto[2], and Anita L. Ganesan[1]

[1]School of Geographical Sciences, University of Bristol, Bristol, United Kingdom
[2]Climate System Research, Finnish Meteorological Institute, 00560 Helsinki, Finland
[3]School of Chemistry, University of Bristol, Bristol, United Kingdom
[4]Department of Engineering Mathematics, University of Bristol, Bristol, United Kingdom

**Correspondence:** Rebecca H. Ward (rebecca.ward@fmi.fi)

**Abstract.**

The Arctic is experiencing unprecedented environmental changes with rapidly rising temperatures. Emissions of methane ($CH_4$) – a potent greenhouse gas – may be increasing from the region, making accurate monitoring essential. The TROPOspheric Monitoring Instrument (TROPOMI) instrument offers high spatial and temporal coverage of $CH_4$ column mole fractions. However, its data in the Arctic has historically exhibited seasonal and latitudinal biases and low-quality retrievals. A major challenge is the lack of ground-based validation data in high-latitude regions, which are used to improve satellite retrievals. This study evaluates inverse modelling to estimate $CH_4$ emissions using TROPOMI measurements over the North Slope of Alaska. Using two retrieval products – the operational SRON product and the scientific WFMD product from the University of Bremen – we assess the alignment of derived emissions with surface measurement-derived inversions over 2018–2020 and test their robustness through sensitivity analyses. Our results show that tundra emissions from SRON inversions align more closely with surface measurement-derived emissions than WFMD inversions. Both TROPOMI-product derived emissions have anomalously low emissions in August 2018 compared to surface measurement-derived emissions, likely due to low data density resulting from high cloud cover. TROPOMI inversions provided stronger constraints on fugitive anthropogenic emissions compared to surface inversions. However, each retrieval produced different emission estimates, highlighting retrieval-dependent differences. Sensitivity tests revealed a strong prior dependence in both retrievals, raising concerns about robustness in northern high latitudes. This study highlights the importance of using multiple retrievals and rigorous sensitivity testing in high-latitude satellite inversions.



## 1 Introduction

Temperatures in the Arctic have been increasing four times the global average (Rantanen et al., 2022), leading to unprecedented environmental changes, including the thawing of permafrost (Box et al., 2019). This thawing has the potential to trigger a positive climate feedback, releasing carbon dioxide ($CO_2$) and methane ($CH_4$) to the atmosphere, and further accelerating warming (IPCC, 2019).

Atmospheric mole fractions of $CH_4$, the second largest anthropogenic contributor to global warming after $CO_2$, are rising

globally (Lan and Dlugokencky, 2022), with $CH_4$ emissions from the Arctic particularly uncertain (Saunois et al., 2020). Regional studies have recently shown an increase in emissions from the tundra region of the North Slope of Alaska, linked to rising temperatures (Sweeney et al., 2016; Ward et al., 2024). However, global studies have not yet demonstrated this increase to be widespread from the whole Arctic region (Lan et al., 2021; Hugelius et al., 2024).

Arctic $CH_4$ emissions during the cold season (September–May) can comprise up to 50% of its annual emissions (Mas-

tepanov et al., 2008; Zona et al., 2016; Kittler et al., 2017; Treat et al., 2018; Ward et al., 2024). This season includes the zero curtain period (September–December) when the active layer of permafrost remains near 0°C below the frozen surface and $CH_4$ production still occurs (Outcalt et al., 1990). However, there are few studies of cold season $CH_4$ emissions despite the importance of the period (Treat et al., 2018).

Emissions of $CH_4$ from the Arctic have previously been estimated using a network of surface mole fraction measurements

in atmospheric inversions (Berchet et al., 2015; Thompson et al., 2017; Ishizawa et al., 2019; Tenkanen et al., 2021; Wittig et al., 2023). Wittig et al. (2024) demonstrated that although the current observation network can detect Arctic $CH_4$ emission trends, its sparse distribution poses significant challenges for accurate quantification and spatial attribution of these emissions. The coverage of this surface network is threatened further by a loss of measurement availability from Russia (Schuur et al., 2024), meaning that measurements from Siberia, with land types that do not have analogues in other regions of the Arctic, will

not be included in future estimates.

Satellite measurements of the column-averaged dry-air $CH_4$ mole fraction ($XCH_4$) offer a potential supplement to surface data in the Arctic. The TROPOspheric Monitoring Instrument (TROPOMI) on the Sentinel 5 Precursor satellite offers high spatial (5 km x 7 km) and temporal (daily revisit) resolutions, presenting possibilities for atmospheric monitoring. TROPOMI measurements are processed using two main retrieval methods: the operational product from the Netherlands Institute for

Space Research (TROPOMI-OPER) (Hu et al., 2016) and the Weighting Function Modified Differential Optical Absorption Spectroscopy (TROPOMI-WFMD) product developed by the University of Bremen (Schneising et al., 2019).

Retrievals from instruments on board satellites such as SCIAMACHY (Buchwitz et al., 2005) and GOSAT (Parker et al., 2011) have facilitated global and regional $CH_4$ inversions (Bergamaschi et al., 2009; Houweling et al., 2014; Alexe et al., 2015; Turner et al., 2015; Pandey et al., 2016; Wang et al., 2019; Maasakkers et al., 2019; Zhang et al., 2021; Lu et al.,

2022). However, few studies have utilised GOSAT or SCIAMACHY for inversions over high latitudes due to challenges from limited and biased measurements, as well as difficulties with vertical profile modelling in these regions, challenges which also apply to TROPOMI. The reliance on sunlight for passive short-wave infrared (SWIR) measurements at a high solar zenith



angle means that there are sparse measurements during the cold season (Turner et al., 2015), limiting the ability to study this potentially high-$CH_4$-emitting period. The steep vertical gradient of $CH_4$ in the stratosphere and the large contribution of the

stratosphere to the total atmospheric column near the poles necessitate accurate stratospheric modelling for precise satellite retrievals (Jacob et al., 2016). Errors in stratospheric $CH_4$ profiles can significantly impact the inferred surface emissions from satellite data (Ostler et al., 2016). The polar vortex, a large area of low pressure and cold air surrounding the Earth's poles, can create distinct vertical $CH_4$ profiles and influence $CH_4$ transport, further complicating the modelling process in high latitudes (Ostler et al., 2014). Limited data, seasonal biases and modelling challenges have previously led to the exclusion of high-

latitude satellite data from inversion studies, including those using TROPOMI measurements (Bergamaschi et al., 2009; Alexe et al., 2015; Houweling et al., 2017; Maasakkers et al., 2019).

There are a small number of inversion studies that use TROPOMI over northern high latitudes, and they reveal persistent seasonal biases (Qu et al., 2021; Tsuruta et al., 2023). Tsuruta et al. (2023) identified ongoing issues with transport model biases in $CH_4$ profiles, of which some have been addressed by recent updates to the TROPOMI-WFMD and TROPOMI-OPER

products (Schneising et al., 2023; Lorente et al., 2021, 2023).

The ongoing validation of TROPOMI data, including comparisons with $XCH_4$ measurements such as the Total Carbon Column Observing Network (TCCON) (Wunch et al., 2011), the Collaborative Carbon Column Observing Network (COCCON) (Frey et al., 2019) or vertical profile measurements from AirCore devices on balloon flights (Degen et al., 2024) and aircraft campaigns (Paris et al., 2014; Narbaud et al., 2023), remains crucial for ensuring accuracy, reliability and the monitoring of

biases (Turner et al., 2019). However, much of the Arctic lacks these supporting measurements, with only three Arctic TCCON stations, four Arctic COCCON stations, and sparse, discrete AirCore balloon and aircraft campaign measurements that are not continuous or long-term, leading to significant gaps in validation datasets for the region. A comprehensive validation of TROPOMI-OPER and TROPOMI-WFMD retrievals over the northern high latitudes found that the most recent versions of these products show significant improvements over previous versions, though some anomalies persist during the cold season

(Lindqvist et al., 2024).

This study investigates the use of TROPOMI data for $CH_4$ flux estimation in the high-latitude Arctic. We compare emissions derived from satellite data, employing both TROPOMI-OPER and TROPOMI-WFMD products, with those derived from surface in situ observations. We emphasise that emissions derived using surface measurements can contain their own biases and uncertainties, however, inversions using surface measurements have been more extensively evaluated in a variety of appli-

cations. We infer emissions using a hierarchical Bayesian Markov Chain Monte Carlo (HBMCMC) inversion over the North Slope of Alaska (NSA) region for 2018-2020. We focus on the NSA due to its high coverage of tundra and continuos permafrost, and because of increasing emissions shown in recent decades (Sweeney et al., 2016; Ward et al., 2024). Compared to other Arctic regions of interest, such as the West Siberian Lowlands, the NSA has much less overlap between anthropogenic and biogenic $CH_4$ sources, reducing uncertainties related to source attribution in the inversion. To evaluate the robustness of the

TROPOMI inversions, we perform sensitivity tests by varying key inversion inputs and assessing their impact on the posterior emissions. Additionally, we conduct inversions combining surface and satellite data.



Sect. 2 introduces the data used in this work and a description of the inversion method and sensitivity tests. Sect. 3 describes the results of the inversions and sensitivity testing. Sect. 4 discusses the results, evaluates of the use of TROPOMI measurements over the NSA and draws comparisons to previous high-latitude TROPOMI inversion studies. Potential improvements to the inversion set-up in this and future high-latitude studies are discussed in Sect. 5.

## 2 Methods

Throughout this work, we define the early season as March–May, and the summertime/growing season as June–August. While the late season has been previously defined as September–December (Ward et al., 2024), we consider only September and October, as the satellite retrievals from TROPOMI over the study region do not extend beyond October.

We primarily focus on emissions from all natural sources in the NSA, referred to here as the tundra sector, due to its regional importance and its potential for increasing $CH_4$ emissions. We also examine anthropogenic emissions, primarily fugitive $CH_4$ from the Prudhoe Bay oil field. In this study the NSA region is defined as 65°N to 72°N and 172°W to 140°W.

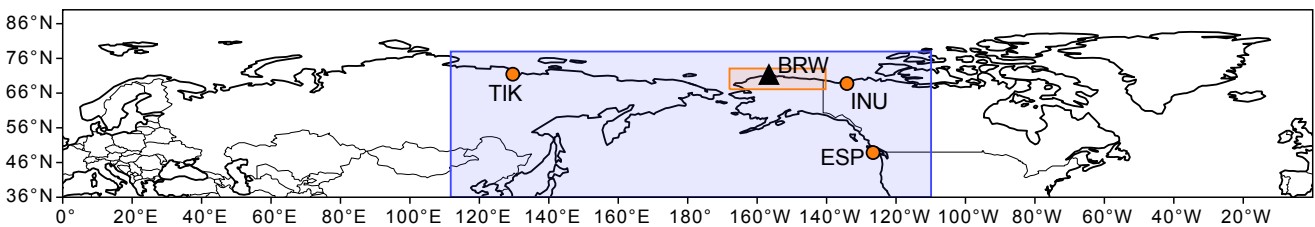

**Figure 1.** Regions, domains and sites used in this study. The location of the NSA region as defined in this study is shown with an orange rectangle. The Arctic inversion domain used in Ward et al. (2024) spans the entire area shown, compared to the smaller Alaska inversion domain (blue rectangle) used in this study. The location of BRW is indicated by a black triangle and the three additional sites TIK, INU and ESP are shown with an orange circle.

### 2.1 Inversion method

The inversions in this work use the same method as Ward et al. (2024). $CH_4$ mole fractions (for surface inversions) or $XCH_4$ column-averaged mole fractions (for satellite inversions), represented by $\mathbf{y}$, are related to a vector of unknown emissions $\mathbf{x}$ and unknown boundary conditions $\mathbf{x_b}$ in the forward model,

$$\mathbf{y} = \mathbf{Hx} + \mathbf{H_b x_b} + \epsilon. \tag{1}$$

Here, $\mathbf{H}$ is a matrix of sensitivities of the measurements to emissions and $\mathbf{H_b}$ is a matrix of sensitivities of the measurements to boundary conditions, produced using the NAME atmospheric transport model (described in Sect. 2.4), with model-measurement errors represented by $\epsilon$.

To estimate the posterior emissions and boundary conditions from $CH_4$ or $XCH_4$ we use a hierarchical Bayesian inversion framework (Ganesan et al., 2014). We estimate monthly scaling factors for both the emissions and for the background $CH_4$





mole fractions of the study domain (the boundary conditions). To avoid non-physical negative emissions, we use a truncated normal prior probability density function (PDF) for both the emissions and boundary conditions. Additionally, we account for

uncertainty in the model-measurement error, treating it as a variable with its own PDF to avoid using a fixed, subjective model error. The model-measurement error accounts for both the known measurement error and unknown model error, combined in quadrature. The PDFs used in the inversions are described in Table 1.

| Variable | Prior Probability Distribution | Parameters |
|---|---|---|
| Emissions scaling factor, $\mathbf{x}$ | Truncated normal (truncated at 0) | $\mu = 1, \sigma = 2$ |
| Boundary conditions scaling factor for surface inversions, $\mathbf{x_b}$ | Truncated normal (truncated at 0) | $\mu = 1, \sigma = 0.02$ |
| Boundary conditions scaling factor for satellite inversions, $\mathbf{x_b}$ | Truncated normal (truncated at 0) | $\mu = 1, \sigma = 1$ |
| Model error | Uniform | 0.5 - 30 ppb |

**Table 1.** Prior probability density functions used for each variable in the HBMCMC method.

We employ a two-stage sampling approach. The emissions and boundary conditions are sampled using a No-U-Turn (NUTS) sampler (Hoffman and Gelman, 2014), while the model-measurement error is sampled using a faster Slice sampler (Neal, 2003).

We set the tuning phase to 1,500 iterations and then sample for 3,000 iterations, discarding the first 20% of samples as burn-in. To ensure convergence, we apply the Gelman-Rubin diagnostic (Gelman and Rubin, 1992) with a criterion of less than 1.05. From the remaining samples, we calculate the mean scaling factor – used to scale the prior fluxes to the posterior fluxes – as well as the associated 95% uncertainty intervals. For example, for total emissions from the NSA, we calculate posterior fluxes for each MCMC sample by multiplying the prior fluxes by each sampled scaling factor and then derive the mean and 95%

uncertainty intervals (in Tg yr$^{-1}$) from the resulting distribution.

In Ward et al. (2024), an "Arctic" inversion spatial domain was used that covered the entire 360° of all latitudes greater than 36°N, selected for its ability to model the transport of particles near the poles, where particles can traverse the entire domain within the model simulation. However, this comes at a higher computational expense, making it less feasible for use with high-density TROPOMI data. This constraint is recognised as a limitation of our current study. For the current study, all

inversions use a smaller "Alaska" domain (36°N to 78°W and 112°E to 110°W). Both domains are shown in Figure 1. The impact of using a different domain on the surface inversion is discussed in Sect. 3.1.

In Ward et al. (2024), the boundary conditions in the North Slope of Alaska inversion were fixed during the inversion, as opposed to allowing variable boundary conditions. Instead a wind direction-derived boundary condition was subtracted from the observations prior to the inversion. A sensitivity analysis conducted as part of Ward et al. (2024) indicated minimal

difference between using fixed and variable boundary conditions. In the present study, we allow variable boundary conditions that we estimate throughout the inversions for both surface and satellite inversions to ensure consistency in the method.

We aggregate grid cells and solve for fewer unknowns in the inversion than the output grid resolution of the transport model. The grid cells are more densely concentrated in areas of interest and more sparsely distributed further from the region of interest, resulting in a total of 297 scaling factors for the emissions for each month. In the satellite inversion, we solve the



boundary condition scaling factors across 4 vertical blocks – spanning 5 km intervals from 0 to 20 km – each divided into the North, East, South, and West directions, resulting in 16 scaling factors. We only use one scaling parameter in each direction (4 total) for surface inversion.

Henceforth, the "main" inversion refers to satellite inversions that use the configuration described in this section, and any changes to this configuration are sensitivity tests (described in Sect. 2.6).

## 2.2 Surface observations

We use high frequency in situ atmospheric mole fraction measurements of $CH_4$ from the National Oceanic and Atmospheric Administration (NOAA) Barrow station (BRW, 71.3° N, 156.6° W) (Dlugokencky et al., 1995). We filter the BRW observations for low wind speeds and those coming from the direction of Utigiavik, a nearby town (see Ward et al., 2024). In this work, we also include a separate inversion using additional measurements from two measurement sites managed by Environment and

Climate Change Canada (ECCC), Inuvik (INU, 68.3° N, 133.5° W) and Estevan Point (ESP, 49.3° N, 126.5° W) (Schuldt et al., 2024), and the Finnish Meteorological Institute (FMI) site Tiksi (TIK, 71.6° N, 128.9° E) (Laurila, 2024). These sites were chosen because they are all within the study domain and have measurements within the study period (2018-2020). The locations of the sites are shown in Figure 1. The measurements from these sites are filtered using a local influence filter described in Ward et al. (2024). The impact of using multiple sites on the surface inversion is discussed in Sect. 3.1.

## 2.3 Satellite observations

We use two satellite retrieval products of column-averaged dry air mole fractions of $CH_4$ ($XCH_4$): the University of Bremen's WFM-DOAS v1.8 (Schneising et al., 2019, 2023) and the SRON operational product v2.04.00 (Lorente et al., 2021, 2023), which we term TROPOMI-WFMD and TROPOMI-OPER, respectively. For both products, the measurement error is taken from the 1-$\sigma$ retrieval uncertainty.

All data have low-quality observations removed using the filters and quality flags recommended by each product, and are re-gridded to 0.234° x 0.352° spatial resolution (the resolution of the atmospheric transport model, see Section 2.4), using a conservative interpolation method, and averaged to an hourly temporal resolution. For both TROPOMI retrievals, we removed all retrievals over the ocean due to a lack of validation data over the ocean globally. We reduced the density of the TROPOMI observations to make it computationally feasible to run our atmospheric transport model. We prioritised the data from our main

region of interest (the NSA) and retained fewer observations from areas outside the NSA but within the inversion domain, which helped to constrain boundary conditions. This resampling involved dividing the domain into regions with varying data retention rates and a random sampler. After regridding, we retained all data over the NSA, 50% from the rest of the state of Alaska, and 5% from the remaining areas. TROPOMI-WFMD observations and observational density after regridding but before and after resampling can be seen for an example month (June 2019) over the Alaska domain in Figure 2. The effects

of these data retention strategies on boundary conditions were evaluated using a sensitivity test described in Sect. 2.6. After resampling, across the whole domain ∼510000 retrievals remain for TROPOMI-OPER and ∼580000 for TROPOMI-WFMD.





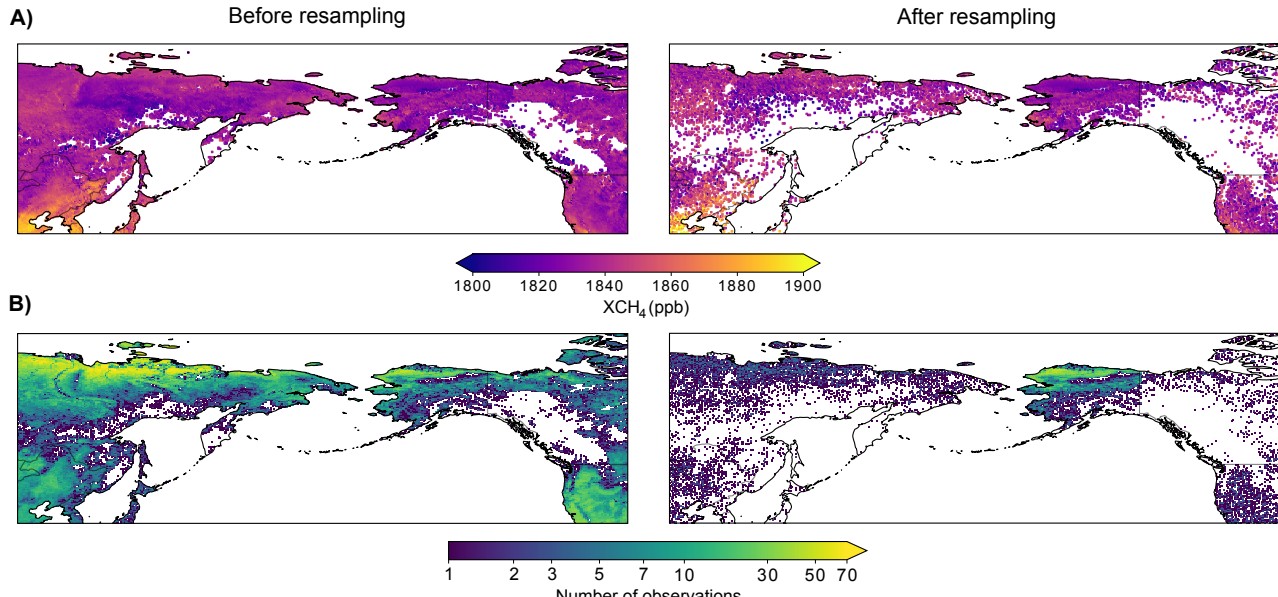

**Figure 2.** Example of observation coverage after regridding from the TROPOMI-WFMD retrieval for June 2019 over the Alaska study domain, before resampling (left) and after resampling (right). The top panel A) shows all observations in June 2019 and the bottom panel B) shows the number of observations (plotted on a logarithmic scale with zero observations shown in white) per $0.234°$ x $0.352°$ grid cell.

Due to high solar zenith angles and low sunlight, there are no observations for either TROPOMI-OPER or TROPOMI-WFMD from late October to early March. However, temporal coverage of TROPOMI-WFMD extends further into March and October than TROPOMI-OPER. As a result, TROPOMI-WFMD may have more potential to constrain emissions during

the cold season. The number of retrievals over the NSA for TROPOMI-OPER and TROPOMI-WFMD (after regridding and resampling) is shown for all years in Figure S1.

The time range for our analysis is from May 2018 to October 2020. TROPOMI's fully functioning operating capacity began in late April 2018, so we do not include data from before this date.

### 2.4 Atmospheric transport model

To provide the sensitivity of XCH$_4$ observations to emissions from the surface we used the Met Office's Numerical Atmospheric-dispersion Modelling Environment (NAME) (Jones et al., 2007), a Lagrangian particle dispersion model (LPDM). We run NAME in backward mode, releasing model particles for each TROPOMI observation, which are tracked backward in time over a 30-day period. We record the sensitivity of the surface emissions (0-40 metres above ground level) and the boundary conditions to the measurements. The NAME output has a resolution of $0.234°$ by $0.352°$. We use the "Alaska" domain shown

in Figure 1.



Meteorological inputs to NAME are provided by the Met Office's Unified Model (Walters et al., 2019), at a spatial resolution of 0.141° x 0.094° and 3-hourly temporal resolution. For each satellite product, we release particles uniformly across different heights defined by the levels within each retrieval. We ran NAME with a maximum altitude of 20 km, which corresponds to maximum level of 11 of 12 for TROPOMI-OPER and 19 of 20 for TROPOMI-WFMD. For the top levels that are above

the maximum level of NAME, the $CH_4$ concentration was assumed to be equal to the corresponding value in the a priori profile from the retrieval. The average footprint sensitivity for TROPOMI (after resampling) are shown in Figure 3 and for the BRW-only inversion and the inversion using multiple surface sites in Figure S2.

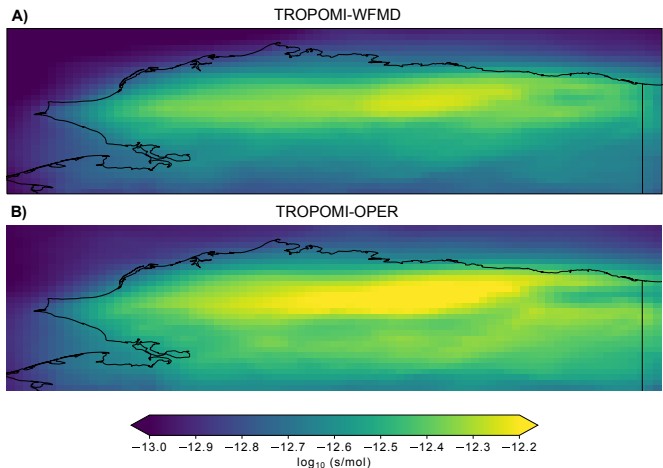

**Figure 3.** Average footprints in 2019 for A) TROPOMI-WFMD after resampling and B) TROPOMI-OPER after resampling over the North Slope of Alaska.

To compare modelled mole fractions to satellite observations, we utilise the methodology derived in O'Dell et al. (2012) and described in Ganesan et al. (2017). We convert our modelled $CH_4$ for each retrieval into a column mole fraction ($XCH_4^{model}$)

for each time index, $t$, as follows,

$$XCH_4^{model}|_t = \sum_1^n p_i \left[ A_i \cdot CH_{4,i}^{model} + (1 - A_i) \cdot CH_{4,i}^{prior} \right], \quad (2)$$

where $p_i$ is the pressure weight at each retrieval level $i$ from the surface to the top of the atmosphere, $A_i$ is the averaging kernel, $CH_{4,i}^{model}$ is the modelled $CH_4$ mole fraction and $CH_{4,i}^{prior}$ is the a priori profile. TROPOMI-WFMD retrievals provide all the variables needed for Eq. 2, but for TROPOMI-OPER we calculate the pressure weights as $p_i = \frac{V_{air,i}}{\sum_i^n V_{air,i}}$, where $V_{air}$ is the dry

air sub-column.

We must adapt Eq. 2 such that we can calculate $\mathbf{XCH_4^{model}}$ in the form of the forward model, Eq. 1, defined in Sect. 3.1. Our forward modelled mole fraction at each level, $i$, ($CH_{4,i}^{model}$ from Eq.2) is due to emissions from the surface within our model domain, and incoming mole fractions from the boundaries,

$$CH_{4,i}^{model}|_t = \mathbf{h_i} \cdot \mathbf{q} + \mathbf{h_{b,i}} \cdot \mathbf{b}, \quad (3)$$



where $\mathbf{h_i}$ is our sensitivity to the surface derived from NAME, $\mathbf{q}$ is our emissions vector, $\mathbf{h_{b,i}}$ is our sensitivity to the boundary conditions and $\mathbf{b}$ is our boundary conditions vector.

$CH_{4,i}^{prior}$, $A_i$ and $p_i$ are provided from the retrieval product, so the second term of Eq. 2 can be subtracted from $XCH_4^{model}|_t$.
Additionally, because NAME is only run up to 20km altitude, we assume the prior profile above this level and we can also

subtract this known contribution from $XCH_4^{model}|_t$. We thus derive a perturbed column-averaged mole fraction, $XCH_{4,pert}^{model}|_t$, to be

$$XCH_{4,pert}^{model}|_t = XCH_4^{model}|_t - \sum_{1}^{maxlev} p_i(1-A_i) \cdot CH_{4,i}^{prior} - \sum_{maxlev}^{n} p_i \cdot CH_{4,i}^{prior}, \tag{4}$$

where maxlev is the maximum level of each retrieval for which the NAME model is run, as described above. A full derivation for Eq. 4 is found in Supplementary Note 2. Substituting Eq. 3 into Eq. 4 gives

$$XCH_{4,pert}^{model}|_t = \sum_{1}^{maxlev} p_i \cdot A_i \cdot (\mathbf{h_i} \cdot \mathbf{q} + \mathbf{h_{b,i}} \cdot \mathbf{b}). \tag{5}$$

Over all times, $t$, Eq. 5 can be written in the form,

$$\mathbf{XCH_{4,pert}^{model}} = \mathbf{Hq} + \mathbf{H_b b}. \tag{6}$$

Here $\mathbf{H}$ and $\mathbf{H_b}$ are matrices of rows t, where each column is the sum of $p_i \cdot A_i \cdot \mathbf{h_i}$ and $p_i \cdot A_i \cdot \mathbf{h_{b,i}}$, respectively. Eq. 6 is now consistent with Eq. 1. In practice, because $\mathbf{q}$ and $\mathbf{b}$ are not dependent on the level, they are stacked into the vector $\mathbf{x}$ such

that,

$$\mathbf{XCH_{4,pert}^{model}} = \mathbf{Hx}. \tag{7}$$

where each corresponding row of $\mathbf{H}$ is a weighted sum over all levels, allowing satellite observations to be used in the same inversion framework as surface measurements.

## 2.5 Prior boundary conditions and emissions

A priori vertical boundary conditions are derived from CAMS v19r1 (Inness et al., 2019). This product uses surface observations in an inversion to produce global atmospheric concentrations. As the product runs only until 2019, we assume that prior boundary conditions for 2020 are the same as in 2019 (discussed in Sec. 5).

Our a priori emissions field comprises emissions from three major sources over the Arctic: natural emissions from tundra, anthropogenic activities, and fires. For the main inversion, we utilise the "Late-season Zona" emissions as detailed in Ward et al.

(2024). For emissions from tundra, we employ a seasonal profile derived from the findings in Zona et al. (2016), complemented by spatial data from the July values of the fractional wetland satellite product SWAMPs (Surface Water Microwave Product Series) (Jensen and Mcdonald, 2019). This approach is designed to allow emissions from areas of less inundated or frozen ground, as documented in Zona et al. (2016). Additionally, we incorporate anthropogenic emissions data from the Emissions





Database for Global Atmospheric Research (EDGAR) v6 (Crippa et al., 2020) and fire emissions data from the Global Fire
Emissions Database (GFED) v4 (Randerson et al., 2017). The a priori emissions are shown in Figure S3. The majority of
emissions over the NSA are from the tundra sector, with significant emissions from the anthropogenic sector over the Prudhoe
Bay oil fields on the Northeast coast. Finally, there are small emissions from fires in the South of the defined region. We do
not include prior emissions from two other important sources in the NHLs: ocean and freshwater (lakes and rivers) emissions.
In Ward et al. (2024), ocean emissions from Weber et al. (2019) were included as a sensitivity test, and their inclusion had
no significant impact on the posterior emissions. For freshwater emissions, there is substantial spatial overlap with tundra
emissions over the NSA, which can lead to double counting if both sources are included separately (Thornton et al., 2016).
Including freshwater emissions in this region in the priors may therefore result in unrealistically high total prior emissions. The
posterior tundra emissions can be interpreted as encompassing both tundra and freshwater methane sources.

To apportion the posterior emissions estimated in each grid cell into one of the three major source sectors, the fraction of
each source in the prior emissions field was used. The distributions of each of these sectors are largely separated spatially as
shown in Figure S4.

## 2.6 Sensitivity tests

Sensitivity tests were conducted to assess the robustness of the derived emissions to changes in the inversion setup. These tests
included changing the prior used in the inversion to the "Uniform" prior from Ward et al. (2024), which uniformly distributes
natural emissions from wetlands and tundra across all months, using the same spatial map as the "Late-season Zona" prior but
setting the emission values to the July level for every month. We tested two different grid cell aggregation formats of 104 and
421 cells (shown in Figure S5). An additional test was carried out using a single scaling parameter for each boundary direction,
such that only one block for each of the North, East, South, and West directions is solved for, in contrast to the main inversion
which uses four blocks in each direction. Finally, to assess the impact of the spatial coverage of observations across the study
250 domain, particularly on the boundary conditions, we removed TROPOMI observations from outside the state of Alaska. A
summary of each sensitivity tests is shown in Table 2.

We additionally carried out an inversion using measurements from both TROPOMI and BRW. Combined TROPOMI and
surface measurement inversions are referred to as TROPOMI-OPER+BRW or TROPOMI-WFMD+BRW throughout the text.
We include an extra offset parameter, with a uniform PDF with a range of 0-10 ppb applied to the BRW measurements, as a
255 method to account for systematic differences between the TROPOMI and BRW observations and in modelling each dataset by
the NAME model.



| Sensitivity test name | Details |
|---|---|
| Uniform prior emissions | Using prior emissions with a constant seasonal profile. Each each month has the same total emissions value. |
| Grid cell aggregations | Using aggregations of 104 or 421 grid cells. 297 in the main inversion. |
| Single BC scaling parameter | Using 1 scaling parameter in each of the N, E, S and W directions rather than 4 vertical blocks. |
| Alaska observations only | Using only TROPOMI observations from within the Alaska rather than from the whole domain. |
| Combined TROPOMI+BRW | Using observations from both TROPOMI and the BRW surface station in a combined inversion. |

**Table 2.** Each of the sensitivity tests described in Sect. 2.6.

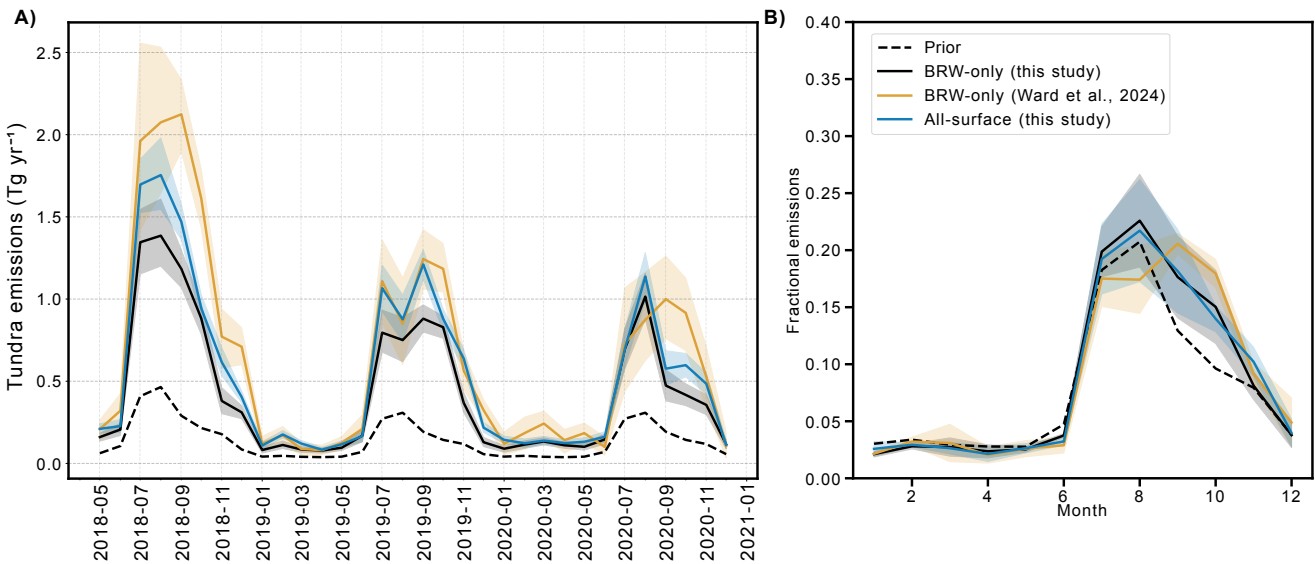

**Figure 4.** Tundra CH$_4$ emissions (A) and seasonal profiles (B) over the North Slope of Alaska (NSA) as derived from the BRW-only inversion for the Alaska domain (this study), in black, the BRW-only inversion for the Arctic domain ((Ward et al., 2024)), in orange, and the all-surface sites inversion for the Alaska domain (this study), in blue.

# 3 Results

## 3.1 Inversion using in situ measurements from surface sites

Here, satellite inversions are compared to a surface inversion using in situ measurements from BRW. We ran satellite and surface-measurement-based inversions over the same Alaska domain to compare them to the surface-measurement-based study in Ward et al. (2024), which used the more spatially extensive Arctic domain.





Figure 4A shows that, in the NSA, the impact of the size of the inversion domains on estimated tundra emissions varies year to year. While emissions for many months are consistent within uncertainties, inversions using the Arctic domain result in higher tundra emissions during the summer and late seasons of 2018, and the late seasons of 2019 and 2020. This suggests that domain size does impact the estimated tundra emissions in this region, potentially due to atmospheric circulation around the poles; however, the general findings using each domain are aligned. We observe significantly higher tundra emissions in our posterior results compared to our prior, including consistent high late-season emissions, a key finding in Ward et al. (2024). Additionally, the average seasonal emission profiles for both domains remain the same within uncertainties, as shown in Figure 4B. While we are satisfied with the similarities between the emissions derived for the NSA using either the Alaska and Arctic domains, further investigation is necessary to fully understand the implications of model domain on emission estimates from the Arctic.

A surface-based inversion was conducted using $CH_4$ measurements from BRW and three additional sites – INU, TIK, and ESP – within the Alaska domain, providing additional constraints outside of the NSA region on derived emissions. For 2020, the "all-surface" derived emissions agrees well with BRW-only (this study) derived emissions, as shown in Figure 4A. However, for the late season months of 2018 and most of 2019, the surface-measurement-derived emissions are larger and closer to the BRW-only (Ward et al., 2024) derived emissions, which use the larger domain. The seasonal emission profile from the surface-measurement-based inversion, shown in Figure 4B, remains consistent with the BRW-only (this study) inversion profile.

For the remainder of this study the BRW-only (this study) inversion using the Alaska domain is referred to as the BRW inversion.

## 3.2 Inversion using measurements from TROPOMI

### 3.2.1 Tundra sector emissions

The tundra emissions derived using the TROPOMI inversions over the NSA exhibit variable agreement with the BRW tundra emissions (outlined in Sect. 3.1). Figure 5A shows the total tundra emissions from the NSA for each month and Figure S6 shows the average (2018–2020) difference maps between the posterior mean and a priori emissions for the early (April–June), summer (July–August) and late (September–October) seasons for BRW, TROPOMI-OPER and TROPOMI-WFMD derived emissions. For most months, TROPOMI-OPER tundra emissions for 2019 and 2020 are largely consistent with BRW tundra emissions within 95% uncertainty intervals. However, there is a notable exception where TROPOMI-OPER falls outside the 95% uncertainty interval, specifically in August 2018. Figure S7 shows the posterior and prior difference maps for July, August and September 2018, where TROPOMI-OPER derived emissions from the tundra region are elevated over the prior in July and September of 2018, but show a decrease in August. This anomaly is discussed in Sec. 4.3.

For most months, TROPOMI-WFMD tundra emissions are significantly lower than BRW-derived emissions within 95% uncertainty intervals. Only in March, April, May and August 2019 and in April, May, June and October 2020 do emissions agree. Otherwise, the TROPOMI-WFMD tundra emissions are less than the BRW-derived tundra emissions. Overall, the TROPOMI-





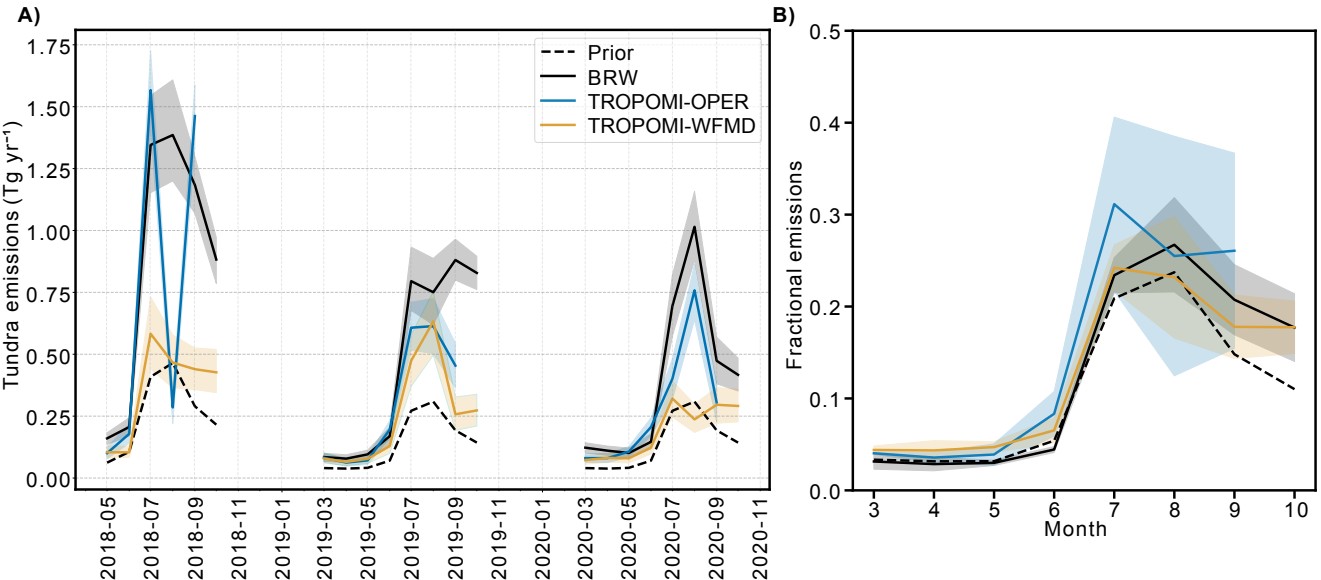

**Figure 5.** Posterior tundra emissions for each of the BRW, TROPOMI-OPER and TROPOMI-WFMD inversions over the NSA. A) shows the monthly emissions over the 2018-2020 period and B) shows average seasonal profile of emissions, expressed as a fraction of annual emissions.

OPER and TROPOMI-WFMD derived tundra emissions agree with the BRW tundra emissions for 15 of 19 months (79%) and 8 of 22 months (36%), respectively.

Comparing the derived emissions of the two TROPOMI retrievals TROPOMI-OPER has higher tundra emissions than TROPOMI-WFMD in May 2018, June 2018, September 2018, September 2019, June 2020 and August 2020. In all other months, the two TROPOMI retrievals are consistent within 95% uncertainty intervals.

**3.2.2 Anthropogenic sector emissions**

There is significant variation in anthropogenic emissions across the different inversions (Figure 6). In May 2018, TROPOMI-OPER estimates of anthropogenic emissions were exceptionally high compared to the TROPOMI-WFMD and BRW estimates. In June, July, and September 2019, TROPOMI-OPER derived anthropogenic emissions increase from the prior, while TROPOMI-WFMD and BRW decrease. Conversely, BRW emissions increase from the prior in August 2019, while TROPOMI-
OPER and TROPOMI-WFMD decrease. In 2020, TROPOMI-OPER derived emissions increase from the prior in March, May, and June; BRW increases in March, August, and October; and TROPOMI-WFMD increases in April and October. Overall, anthropogenic emissions derived from TROPOMI-OPER agree with BRW for 8 of 19 months (42%), while TROPOMI-WFMD agree with BRW for 17 of 22 months (77%). Anthropogenic emissions in the NSA are predominantly from the Prudhoe Bay oil fields. The emissions maps (Figure S6) show that in the March-May emissions from the Prudhoe Bay oil fields are decreased
for both the BRW and TROPOMI-WFMD inversions, but increased for TROPOMI-OPER.





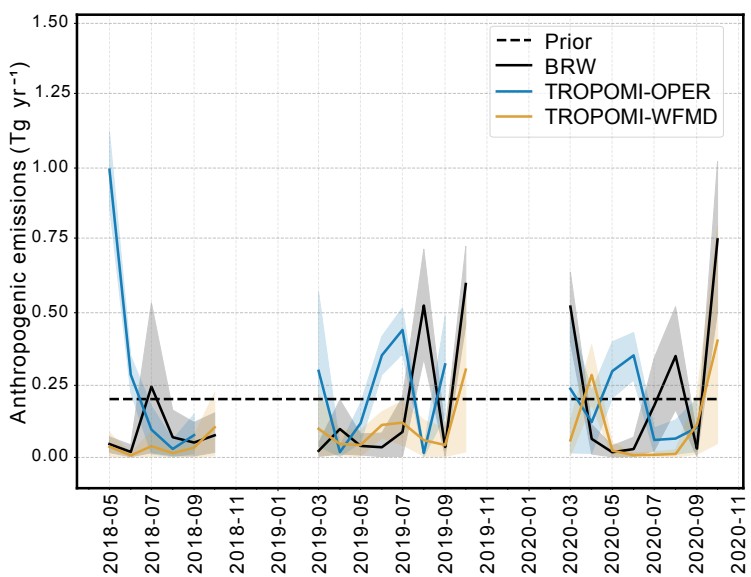

**Figure 6.** Posterior anthropogenic emissions for each of the BRW, TROPOMI-OPER and TROPOMI-WFMD inversions over the NSA over the 2018-2020 period.





## 3.3 Sensitivity tests of TROPOMI inversions



**Figure 7.** Sensitivity tests for the satellite-based inversions. A) and B) show derived tundra emissions from TROPOMI-OPER and TROPOMI-WFMD, respectively. C) and D) show derived anthropogenic emissions from the same inversions. Uncertainty estimates for the sensitivity tests are included in Figures S8-S12.

To assess the robustness of the satellite products to varying inversion inputs and prior choices, we carried out four sensitivity tests, as detailed in Table 2, and observed the impact on the derived tundra and anthropogenic emissions. The results of each sensitivity test are described in the following sections, with the results in Sect. 3.2 referred to as the "Main" inversion. The

monthly total emissions for all tests are shown in Figure 7. We consider the monthly emissions estimate unchanged if the 95%





uncertainty intervals of the derived emissions from the sensitivity test and the main inversion overlap. March–September 2019 and 2020 total budgets are shown in Table S2.

### 3.3.1 Uniform prior emissions

A sensitivity test was conducted to assess the impact of changing seasonality of the prior emissions. For this test, the Late-
Season Zona prior from the main inversion was replaced with a time-invariant (referred to as "Uniform") prior for the tundra sector, as described in Sect. 2.5. This alternative prior configuration allows for emissions during both the early and late seasons. Surface inversions in Ward et al. (2024) showed that the derived emissions were not sensitive to the prior seasonal profile, as the use of the Uniform prior did not change the average posterior seasonality over the period 2000-2020.

The sensitivity test was repeated for the surface inversions detailed in Sect. 3.1. As shown in Figure S8, both the BRW-only
inversion and the all-surface inversion show some sensitivity to the change in prior seasonal profile. The posterior seasonal profile shifts from the Late-Season Zona inversion to the Uniform inversion, indicating a sensitivity to seasonal profile shape that did not occur in Ward et al. (2024). However, a clear seasonal profile, which peaks in the summer and is lowest in the winter, remains visible. This indicates a relatively low sensitivity to prior seasonal profile compared to the satellite inversions, described below.

Tundra emissions from TROPOMI-OPER and TROPOMI-WFMD inversions show a high sensitivity to the prior emissions seasonal profile shape. Posterior tundra emissions (Figure S9) and seasonal profiles (Figure 8A and B), are significantly different between the Late-Season Zona and Uniform priors, and a clear seasonal cycle is no longer visible in the Uniform prior case.

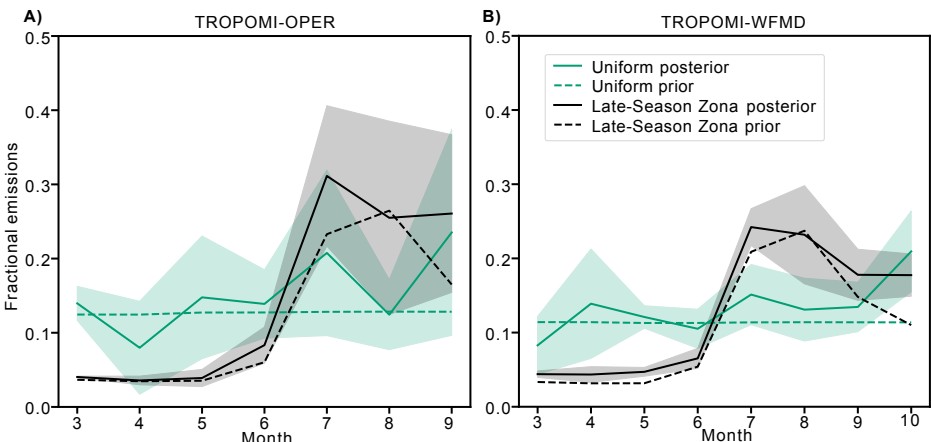

**Figure 8.** Uniform prior sensitivity test for the TROPOMI inversions. Average seasonal profiles of derived tundra emissions for A) TROPOMI-OPER and B) TROPOMI-WFMD for the Uniform prior (green) and Late-Season Zona prior (black). Note the Late-Season Zona prior is the same as the main inversion.





Changing the tundra sector emissions to the Uniform prior had no impact on the posterior emissions from the anthropogenic
sector for any of the inversions.

### 3.3.2 Grid cell aggregations

As described in Sect. 2.1, we aggregate the grid cells of the transport model output in the inversion to solve for fewer parameters
than its native grid resolution. In this sensitivity test, we used varying grid cell aggregations — 104 cells and 421 cells,
compared to the 297 cells used in the main inversion (Figure S5). Comparisons to the main inversion are shown in Figure S10,
while spatial patterns of the resulting scaling factors for selected months are shown in Figure S11.

For the derived tundra emissions, changing the number of grid cells did not result in significant alterations in the monthly
emissions time series within 95% uncertainty intervals for TROPOMI-OPER, apart from in September 2018. For TROPOMI-
WFMD, the summer months of 2018 and 2020 showed significant differences in the 95% uncertainty intervals between the
104 and 421 cell aggregations. The derived anthropogenic emissions were unaffected by the aggregation level for TROPOMI-
WFMD. The TROPOMI-OPER inversion showed only changes for June 2019, where the 104 cell aggregation produced lower
emissions than the 297- and 421-cell aggregations.

Changes in total monthly emissions are limited to a few months. For those months, there are large differences in how the
prior emissions are spatially scaled across the NSA. For example, in the TROPOMI-OPER inversion for September 2018, the
coarser 104-cell aggregation results in a large region of tundra being scaled up uniformly, leading to a noticeable increase in
total emissions. The 297- and 421-cell aggregations allow for smaller areas to be adjusted, concentrating the scaling in fewer
areas and limiting the impact on total emissions. In June 2019, scaling factors over the ocean in the TROPOMI-OPER inversion
show substantial differences across the three different aggregations, suggesting that the treatment of these regions are sensitive
to grid resolution.

Overall, TROPOMI-OPER inversions show greater spatial variability in scaling factors compared to TROPOMI-WFMD.
For example, in August 2020 the TROPOMI-WFMD scaling factors remain relatively homogeneous, whereas TROPOMI-
OPER shows more pronounced spatial heterogeneity, despite only TROPOMI-WFMD showing a change in the overall tundra
emissions for this month.

### 3.3.3 Single boundary condition scaling parameter

We simplified the scaling factors for the boundary conditions compared to the main inversion by solving for a single scaling
parameter for each of the North, East, South, and West boundaries, instead of the four vertical blocks in the main inversion. This
tested whether four vertical blocks accounted for potential inaccuracies in modelling of the boundary conditions, by allowing
for adjustments to the vertical profiles at the boundaries. Comparisons to the main inversion are shown in Figure S12.

For the derived tundra emissions, using a single vertical block resulted in a decrease in derived emissions but the decrease
is not outside of the 95% uncertainty intervals for either inversion. For the derived anthropogenic emissions, when going from
four to one vertical block, the TROPOMI-OPER emissions decreased in June 2018 and June 2019 and the TROPOMI-WFMD
emissions decreased in June 2019. This suggests that allowing for vertical variability in the boundary conditions during the




inversion has a small but statistically insignificant impact on the tundra emissions, and a small impact on the anthropogenic emissions.

### 3.3.4 Observations only over Alaska

To assess the impact on derived emissions and boundary conditions of keeping observations across the whole domain, we removed TROPOMI data from anywhere outside of Alaska. Comparisons to the main inversion are shown in Figure S13.

For the derived tundra emissions, using observations from Alaska resulted in increased emissions estimates outside of 95% uncertainty intervals in the summer months for both TROPOMI-OPER and TROPOMI-WFMD. For the derived anthropogenic emissions, using observations from Alaska only decreased emissions in May 2020 for TROPOMI-OPER and decreased emis-
sions in May 2018 and April 2020 for TROPOMI-WFMD.

With Alaska measurements only, we observed lower boundary conditions. The exclusion of observations near the domain boundaries likely reduced the constraint on the boundary conditions and may have caused the model to underestimate influences from outside the inversion domain, contributing to the observed lowering of the boundary values.

### 3.4 Forward modelling of Barrow mole fractions with TROPOMI derived emissions

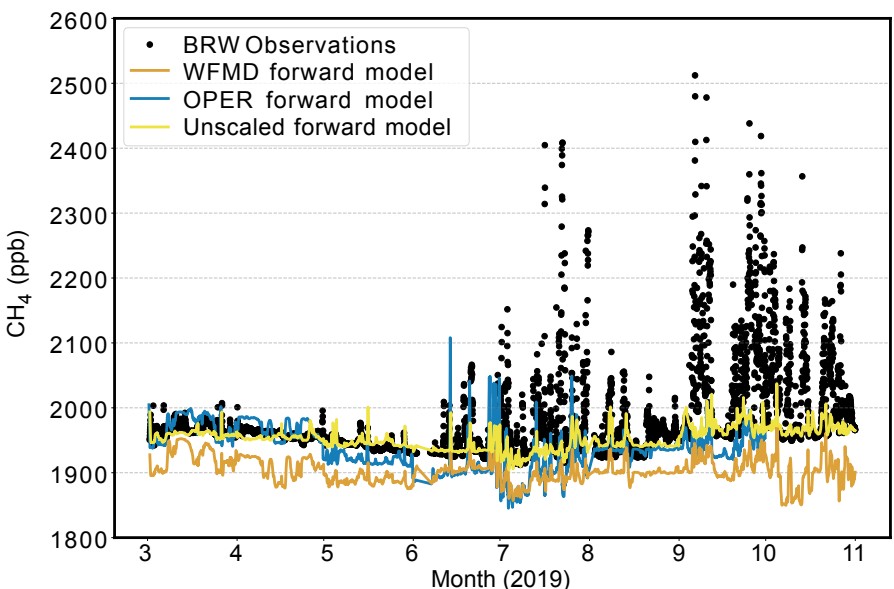

**Figure 9.** Forward modelled mole fractions at BRW for 2019. The forward model using TROPOMI derived emissions and boundary condition scaling factors are shown in blue for TROPOMI-OPER (blue) and TROPOMI-WFMD (orange). The forward model using the prior is shown in yellow and the BRW observed mole fractions are shown by black dots.





To validate the TROPOMI-derived emissions, we forward modelled the TROPOMI posterior emissions to generate modelled mole fractions at BRW, as BRW observations were not assimilated in these inversions. The a priori emissions and boundary conditions, $\mathbf{x}$ and $\mathbf{x_b}$, from the forward model in Equation 1 were scaled by the mean posterior emissions scaling factors and posterior boundary condition scaling factors, derived in the TROPOMI inversions. We compared these scaled forward model estimates to the prior forward model outputs.

Our results indicate that the TROPOMI scaled forward model provides a poorer fit to the BRW observations than the unscaled prior forward model, as shown for 2019 in Figure 9. For the TROPOMI scaled forward model, the baseline is consistently negatively offset for TROPOMI-WFMD (Figure 9B) and negatively offset in the summer months and positively offset in the early season months for TROPOMI-OPER (Figure 9A).

Additionally, forward models incorporating two of the sensitivity tests from Sect. 3.3 are shown in Figure S14 for August
2019: one vertical block boundary conditions, and Alaska observations only. The one vertical block test increases the baseline closer to the observations for TROPOMI-WFMD. However, for TROPOMI-OPER the baseline decreases further from the observations. Limiting the measurement dataset to observations over Alaska increased the baseline for TROPOMI-OPER. For TROPOMI-WFMD, limiting measurements to Alaska only produced an anomalous trough in the baseline around 2019-08-15, which requires further investigation.

**3.5   Combined TROPOMI and Barrow inversion**

We conducted combined TROPOMI and surface measurement inversions, incorporating an additional offset parameter to our hierarchical model to account for any potential monthly biases between modelled satellite and surface mole fractions due to calibration differences, as detailed in Sect. 2.6.

For August 2019, the mean inferred offset was 37 ppb between BRW and TROPOMI-WFMD, and 6 ppb between BRW
and TROPOMI-OPER. These values closely match the differences between the TROPOMI-scaled forward model and BRW observations shown in Figure S14A for the same month, indicating that the offset parameter effectively corrects for this monthly bias.

For the TROPOMI-OPER+BRW inversion, we observed the same derived tundra emissions to the TROPOMI-OPER inversion within 95% uncertainty intervals for most months. However, in September 2019, the TROPOMI-OPER+BRW derived
tundra emissions increased compared to the BRW inversion and TROPOMI-OPER inversion outside of the 95% uncertainty intervals (Figure 10A). The TROPOMI-WFMD+BRW derived tundra emissions were greater than TROPOMI-WFMD derived tundra emissions, and closer to BRW derived tundra emissions for the summer and late season months of 2018, the late season months of 2019 and August 2020 (Figure 10B). Figure S6 shows the average seasonal difference between the posterior and prior emissions from the tundra region for the TROPOMI-OPER+BRW. which are spatially similar to those from the
TROPOMI-OPER inversion. For the TROPOMI-WFMD+BRW inversion, emissions from the tundra region are larger than the prior, whereas emissions from the TROPOMI-WFMD inversion are smaller across the tundra region.

The anthropogenic tundra emissions are shown in Figure S15. In May 2018, TROPOMI-OPER had exceptionally high emissions. However, in the TROPOMI-OPER+BRW the emissions are substantially lower and align with the prior emissions.




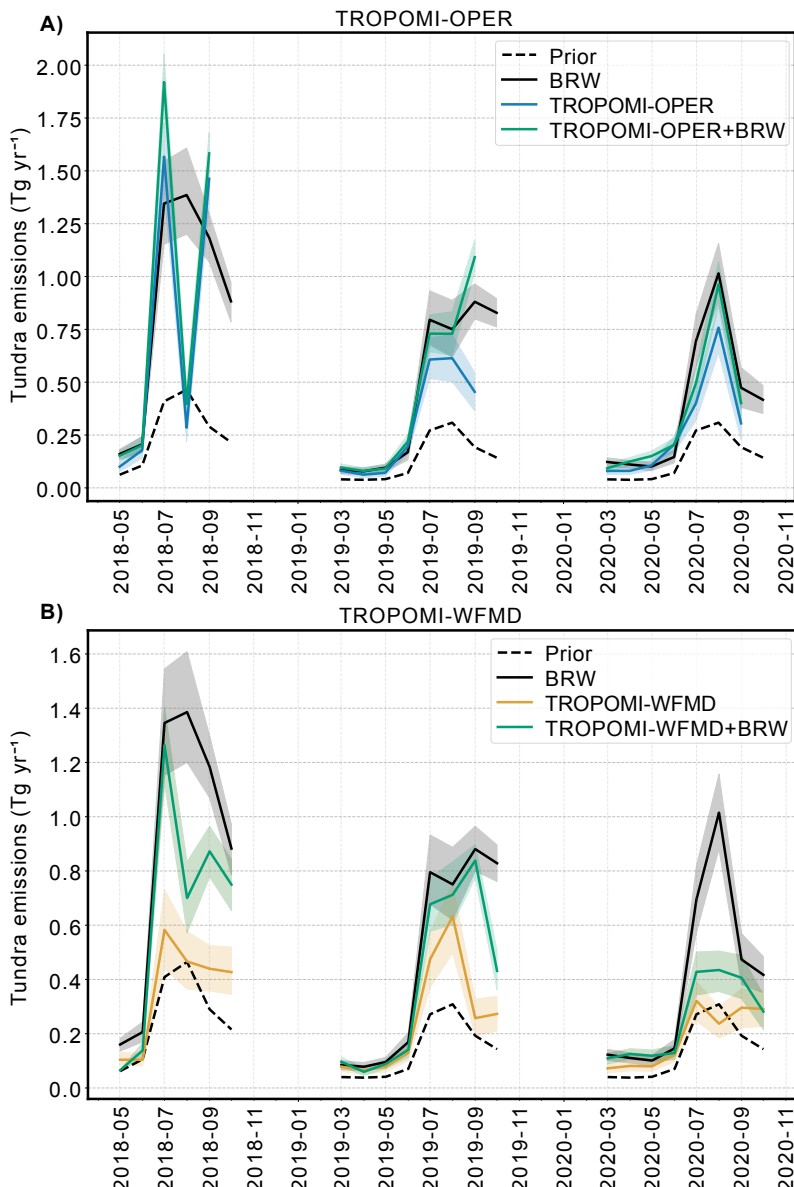

**Figure 10.** Combined TROPOMI+BRW inversions. A) and B) show the tundra sector emissions for the TROPOMI-OPER+BRW and TROPOMI-WFMD+BRW inversions, respectively.

Otherwise, all other months remain consistent with the TROPOMI-OPER only anthropogenic emissions, apart from June 2020
where the emissions decrease from the prior. For the TROPOMI-WFMD+BRW inversion, the derived anthropogenic emissions do not change from the TROPOMI-WFMD only inversion apart from May 2018, March 2020 and August 2020.





### 3.6 Fit of posterior modelled XCH$_4$ to measured XCH$_4$

The root mean square error (RMSE) between the modelled and measured CH$_4$ mole fractions is used to assess the fit of the model to the observations. Table S1 shows the RMSEs for TROPOMI observations, and Table S2 for BRW observations, for each inversion in this study. Both prior (before inference) and posterior (after inference) RMSEs are reported. A lowering of the RMSE from the prior to the posterior for the same observations indicates a better fit, though it does not account for potential overfitting.

Across all inversions, the RMSE decreases from prior to posterior for both TROPOMI and BRW, indicating improved agreement with the observations. TROPOMI-WFMD has consistently lower posterior RMSEs than TROPOMI-OPER, but also consistently higher prior RMSEs.

Changes in RMSE across the sensitivity tests reflect differences in the number of adjustable parameters and observations. For instance, the Alaska-only inversion has fewer observations, resulting in lower RMSEs. The 421 grid cells aggregation results in lower RMSEs than the main inversion due to increased spatial flexibility, allowing more grid cells to be optimised. In contrast, the single vertical block test reduces the degrees of freedom for adjusting boundary conditions, leading to a higher RMSE.

For BRW, the BRW-only Alaska domain inversion has the lowest posterior RMSE. The larger Arctic domain inversion, and the multiple sites and combined TROPOMI inversion, increases the RMSE.

### 3.7 Comparison of inversion uncertainties

In our inversion method, the TROPOMI model-measurement error is a quadrature combination of the measurement error taken from the retrieval and an estimated model error (see Sect. 2.1). Figures S15–S17 show the monthly average measurement error, inferred model error (mean of all MCMC samples), and combined model-measurement error for BRW, TROPOMI-OPER, and TROPOMI-WFMD, respectively.

Over 2018–2020, the retrieval-based measurement error is much lower for TROPOMI-OPER (3 ppb; 1–6 ppb) than for TROPOMI-WFMD (15 ppb; 10–19 ppb; the reported values are the mean error over the full study period, with the numbers in parentheses representing the mean lower and upper bounds of the monthly 95% uncertainty intervals). The inferred model error is higher for TROPOMI-OPER (16 ppb; 13–21 ppb) than for TROPOMI-WFMD (5 ppb; 2–10 ppb). These differences lead to similar combined model-measurement errors for both products: 17 ppb (14–20 ppb) for TROPOMI-OPER and 16 ppb (11–21 ppb) for TROPOMI-WFMD. The combined model-measurement error for both TROPOMI products varies seasonally, with higher uncertainties during the early and late seasons compared to summer.

For the BRW-only inversions, the average model-measurement error is 13 ppb (5–25 ppb). When BRW is combined with TROPOMI, the model-measurement error increases to 18 ppb (9–32 ppb) in the TROPOMI-WFMD+BRW inversion and to 15 ppb (7–28 ppb) in the TROPOMI-OPER+BRW inversion. Across all BRW inversions, the model error component dominates the model-measurement error, and seasonal variation is less pronounced than using TROPOMI.





## 4 Discussion

This section evaluates the use of TROPOMI-derived $CH_4$ emissions for high-latitude inversions, with a particular focus on their applicability over the NSA. Sect. 4.1 compares the TROPOMI inversions with surface-based inversions. Sect. 4.2 evaluates the sensitivity of derived emissions to prior emissions and inversion set-up. Sect. 4.3 discusses the challenges of using TROPOMI products in this region. Sect 4.4 discusses cold season TROPOMI coverage. Finally, sect 4.5 compares inversions in this study to previous TROPOMI inversions over the Arctic.

### 4.1 Emissions derived from TROPOMI and surface-based inversions

TROPOMI-derived emissions can be compared with emissions derived from surface inversions to assess whether different inversion approaches yield consistent results. Here, TROPOMI-derived emissions are considered consistent with surface inversion if they align with the BRW-derived emissions within 95% uncertainty intervals. However, as discussed in Sect. 2, the surface inversions are themselves sensitive to some aspects of the inversion set-up and inputs. Despite the limitations, in situ
measurement-based inversions are a well-established method for emissions quantification in the Arctic and are valuable for comparison with TROPOMI inversions.

The main inversion in Sect. 3.2 shows that the TROPOMI-OPER tundra emissions are consistent with BRW-derived emissions for multiple months and exhibit similar spatial emission patterns. In contrast, TROPOMI-WFMD exhibits different spatial patterns and shows less consistency with BRW-derived emissions. While TROPOMI-OPER demonstrates greater consistency
during the early season and summer, its coverage ends in September, missing the potentially high-emitting late-season period (Zona et al., 2016; Ward et al., 2024). In the combined inversions the TROPOMI-WFMD product shows improved agreement with BRW derived tundra emissions when combined with the BRW data, as shown in Sect. 3.5. For TROPOMI-OPER, combining the BRW data results in minimal changes to derived tundra emissions, though spatial emissions patterns more closely resemble those from the BRW only inversion. For both TROPOMI products, forward modelling (Sect. 3.4) indicates that com-
bined inversions require an offset to account for bias between satellite and surface data. This is effectively handled using an offset parameter in 3.5.

For the anthropogenic sector, the main inversion (Sect. 3.2) indicates that TROPOMI-WFMD derived emissions are more consistent with BRW derived emissions than TROPOMI-OPER, reversing the pattern observed for tundra emissions. The largest anthropogenic emissions in the NSA originate from fugitive releases at the Prudhoe Bay oil fields, where methane
is emitted as discrete point sources rather than the widespread, diffuse emissions from tundra. The BRW station is situated approximately 300 km north-west from Prudhoe Bay, and measurements are less sensitive to emissions from this source compared to the tundra south of the station (this can be seen from the footprint maps in Figure S2). In contrast, TROPOMI provides data points over Prudhoe Bay, offering stronger constraints when clear-sky observations are available. This is confirmed by the combined inversions, where adding BRW data to the TROPOMI-only inversions caused minimal changes to anthropogenic
emissions, indicating TROPOMI provides the dominant constraint. However, despite more coverage over Prudhoe Bay oil





fields, TROPOMI-WFMD and TROPOMI-OPER derived different anthropogenic emissions many months, indicating differences driven by the retrievals.

The intra-annual variability of anthropogenic emissions is unclear from our results. The prior (EDGAR v6) has no monthly variation. However, all inversions in this study show some degree of monthly variability, suggesting dynamics not captured by
the inventory. Hu et al. (2025) identified a winter peak in oil and gas methane emissions in the contiguous U.S., likely linked to higher extraction and transmission rates during the heating season. State-level activity data from the U.S. Energy Information Administration show similar trends for Alaska, with elevated natural gas withdrawals during the cold season, including a notable spike in May 2018 – matching the high emissions inferred by TROPOMI-OPER that month (U.S. Energy Information Administration, 2025). TROPOMI-WFMD aligns most closely with these seasonal activity patterns, while TROPOMI-OPER
and BRW show elevated emissions during summer months, which may reflect overlap with natural emissions from tundra and wetlands near Prudhoe Bay. Additionally, fugitive methane leaks can occur year-round, and activity data alone does not indicate leak magnitude or emission rates. Finally, in terms of emissions magnitude, updates in EDGAR 2024 (Crippa et al., 2024) suggest a ∼60% reduction in emissions totals over Alaska compared to EDGAR v6. If this revised estimate is more accurate, it implies that the prior total used here may overstate actual emissions totals.

### 4.2   Sensitivity of derived emissions to priors and inversion set-up

To assess the robustness of TROPOMI-based inversions, sensitivity tests varying the inversion setup were conducted (Sect. 3.3). Derived emissions are considered robust if they do not change when the inversion set up changes, meaning that the results are stable and reliable across different modelling configurations. The stability would indicate that the inversion method and TROPOMI data can consistently produce accurate emissions estimates despite variations in the setup, thereby enhancing
the confidence in the derived emissions. TROPOMI derived tundra emissions showed significant changes with alterations to the prior seasonal cycle. Both the early season (March–May) and late season (September–October) periods were especially affected when the prior emissions were given a uniform seasonal profile with unrealistically high emissions during the cold season (see Table 2). This highlights the importance of using a well-informed prior, especially in the cold season, consistent with findings by Tenkanen et al. (2021). However, the BRW inversion was much less sensitive to the uniform prior and
still produced plausible posterior emissions despite the unrealistic emissions in the cold season. Overall, the strong dependence of TROPOMI-derived tundra emissions on the prior suggests a lack of robustness, a concern that could extend to other high-latitude Arctic regions. Tests involving grid cell aggregations, boundary condition vertical blocks, and Alaska observations showed smaller or insignificant changes in derived tundra emissions. However, the grid cell aggregations sensitivity test showed spatial differences, with TROPOMI-OPER showing more spatial variation than TROPOMI-WFMD. For anthropogenic emissions, all sensitivity tests resulted in changes in TROPOMI-derived emissions for very few months. Unlike for
pogenic emissions, all sensitivity tests resulted in changes in TROPOMI-derived emissions for very few months. Unlike for tundra emissions, no sensitivity test directly altered the prior anthropogenic emissions, meaning their sensitivity to prior inputs was not assessed, and this should be included in future work.

An important consideration is how observations from outside the NSA influence derived emissions within it. For the surface inversions, excluding sites outside Alaska (i.e. using only BRW on the same domain) resulted in lower tundra emissions





compared to the inversion using more sites. In contrast, for the TROPOMI inversions, restricting observations to Alaska alone, rather than the whole inversion domain, led to increased tundra emissions in some summer months, accompanied by a change in the optimised boundary conditions relative to the prior boundary conditions. Although these setups are not directly comparable, these contrasting results suggest that information from outside the region of interest plays different roles in the surface and satellite inversions.

## 4.3 TROPOMI retrieval and coverage limitations over the NSA

Data providers have corrected for bias in both TROPOMI products used in this study to improve accuracy at high latitudes, and the retrievals have been validated using $XCH_4$ measurements from TCCON sites (Lorente et al., 2021; Schneising et al., 2019; Lorente et al., 2023; Schneising et al., 2023). However, the geographic locations of TCCON, COCCON and AirCore sites relative to the NSA pose significant limitations for bias correction. The nearest TCCON site, Eureka in Nunavut, Canada, is about
2000 km away and the nearest AirCore measurements are in Europe. The closest ground-based column $XCH_4$ measurements come from a COCCON station in Fairbanks, Alaska, located approximately 300 km south of the NSA, and COCCON stations are not currently used for the correction or validation of either of the TROPOMI products used in this study. As Lorente et al. (2023) notes, anomalies have been detected in the TROPOMI-OPER product that are not near to TCCON stations, which is likely to affect the TROPOMI-WFMD product. Consequently, despite the presence of a few ground-based measurement sta-
tions in the Arctic, potential improvements or anomalies over the NSA, as well as other high-latitude regions such as the East Siberian Lowlands and the Taymyr Peninsula (Ward et al., 2024), could be overlooked.

Compared to previous estimates of emissions in the Arctic in August 2018 which typically has high emissions (Tenkanen et al., 2021; Wittig et al., 2023; Tsuruta et al., 2023), a pronounced trough is observed in TROPOMI-OPER tundra emissions over the NSA during August 2018. August is a month of low data coverage over the NSA (Figure S1), and this is likely
due to persistent high cloud cover. Both TROPOMI-OPER and TROPOMI-WFMD implement cloud filtering using data from the Visible Infrared Imaging Radiometer Suite (VIIRS) onboard the Suomi-NPP satellite (Hutchison and Cracknell, 2005), to ensure high-quality retrievals under clear-sky conditions. During the 2018–2020 study period, August 2018 experienced the highest cloud coverage, with clouds present during 88% of daylight hours (Figure S19). The sparse data in August 2018 likely results in the inversion adhering closely to the prior in this month, producing lower emissions estimates than in surrounding
months due to insufficient observational constraint. This underlines the need to continue and expand in situ measurements in important but persistently cloudy methane emitting regions. Future satellite missions, such as the Franco-German Methane Remote Sensing LIDAR Mission (MERLIN) (Ehret et al., 2017), planned to launch in 2028, will potentially have higher data density in cloudy scenes, as the high resolution LIDAR beam will reach the surface through gaps in optically thick clouds.

Despite lower reported measurement errors and prior RMSE, TROPOMI-OPER consistently results in higher posterior
RMSE and larger inferred model error compared to TROPOMI-WFMD. This discrepancy might indicate that TROPOMI-OPER measurement errors are underestimated, leading the inversion to assign more of the model-measurement error to model errors over measurement errors. The greater spatial and temporal variability observed in TROPOMI-OPER scaling factors and monthly emissions estimates suggests it contains finer-scale variations that are more challenging for the inversion to resolve. In



contrast, TROPOMI-WFMDs smoother spatial patterns and higher retrieval uncertainties may provide the inversion with more
flexibility to fit the observations, resulting in lower posterior RMSE and smaller model error estimates.

### 4.4   Cold season coverage improvements and remaining challenges

The cold season in the Arctic tundra has been shown to be a larger contributor to annual $CH_4$ emissions than previously thought
(Mastepanov et al., 2008; Zona et al., 2016; Kittler et al., 2017; Treat et al., 2018). Compared to previous satellite products,
the increased data coverage of TROPOMI during this period, capturing high-resolution measurements at the start of the late
season and end of the early season, potentially enables improved monitoring of this understudied season. This is especially
valuable given the practical challenges of deploying additional in situ measurement sites due to harsh cold season conditions
and the remoteness of Arctic tundra. The main inversions in this study found that after imposing prior tundra emissions with
higher late season emissions derived from Zona et al. (2016), the posterior tundra emissions in September–October increased
further from the prior, aligning with these previous studies that find unexpectedly large emissions from the cold season.
However, despite additional data coverage, the tundra emissions derived from the TROPOMI inversions demonstrated a
lack of robustness in the early and late seasons. This lack of robustness may stem from several factors, including issues
with $XCH_4$ retrievals during these periods due to higher solar zenith angles, albedo effects from snow cover, and seasonal
stratospheric influences, such as the polar vortex. These challenges are reflected in the model errors derived in the inversions. In
the TROPOMI inversions, the model errors are consistently higher during the early and late seasons compared to the growing
season (Figures S16 and S17). In contrast, BRW model errors show year-to-year variability when model errors are highest
(Figure S16) – not consistently in the cold season. This indicates that there may be a systematic difference in how well the
transport model captures boundary-layer versus upper-atmosphere dynamics.

### 4.5   Comparison to other northern high-latitude TROPOMI inversions

Tsuruta et al. (2023) conducted inversions using previous versions of the TROPOMI-WFMD (v1.2) and TROPOMI-OPER
(v01.02.02 and v01.03.01) products for 2018 to assess the utility of TROPOMI products at high latitudes. Unlike our study,
which concentrates on the NSA, their research divides the analysis into larger geographical areas circumpolar north of 45∘N.
Their inversion uses the CTE-$CH_4$ inverse model, which employs an ensemble Kalman filter and an Eulerian atmospheric
transport model, which significantly differs from that used here.
Thompson et al. (2025) recently applied a different LPDM, FLEXPART, to estimate methane emissions from the West
Siberian Lowlands – another key northern high-latitude source region. To reduce computational costs, they developed a re-
trieval averaging routine that aggregates satellite observations onto a coarser spatial grid, based on the standard deviation of the
original observations. This contrasts with our observation filtering approach, which prioritised measurements over our specific
region of interest. However, their inversion domain is considerably smaller than ours, meaning that a larger proportion of avail-
able observations are relevant to their domain by default. The inversion system used by Thompson et al. (2025), FLEXINVERT,
is a variational inversion method that uses a conjugate gradient minimisation with the FLEXPART LPDM.



A direct comparison of the derived emissions derived here and those from Tsuruta et al. (2023) and Thompson et al. (2025) is not possible due to differing geographical focusses. However, similar to Tsuruta et al. (2023), we find that TROPOMI-derived tundra (or natural in Tsuruta et al. (2023)) emissions are lower than those from in situ site-based inversions. Notably, our results show larger deviations from the prior than both Tsuruta et al. (2023) and Thompson et al. (2025), who report posterior emissions more closely aligned with their prior estimates. This discrepancy may reflect differences in the accuracy of the a priori emissions, or the relative strength of prior constraints in the inversion frameworks, which can limit how much the posterior can deviate from the prior.

The treatment of model–measurement errors differs between our inversion, Tsuruta et al. (2023) and Thompson et al. (2025), yet they all converge on similar total model–measurement uncertainties, with Tsuruta et al. (2023) imposing the highest minimum error. Tsuruta et al. (2023) fix their transport model error at 15 ppb and impose a 5 ppb minimum retrieval error (total $\geq$ 20 ppb). Thompson et al. (2025) prescribe combined model–measurement errors of 14–20 ppb based on retrieval errors and an estimate of background uncertainty. In contrast, our hierarchical inversion treats the model–measurement error as a hyperparameter that is inferred to reduce subjective assumptions and values fall in the 11–21 ppb range (Sec. 3.7).

In all three studies, it is assumed that the model-measurement errors are uncorrelated. For future TROPOMI inversions, Tsuruta et al. (2023) recommended increasing model-measurement uncertainty or accounting for model-measurement error correlation in both space and time. The HBMCMC method in this work assumes that uncertainties are uncorrelated both spatially and temporally. In reality, model-measurement errors in satellite data could be highly spatially correlated. From the transport model side, systematic biases in meteorological input data, such as wind speed and direction, can affect multiple measurements in the same area. Additionally, simplified parameterisation of fine-scale processes like turbulence and convection can lead to spatially correlated errors. For satellite measurements, retrievals in close proximity could be impacted by surface albedo biases, such as from snow cover. Calibration issues, such as instrument calibration drifts and inconsistent calibration corrections, can introduce systematic biases over large areas. Data processing, such as spatial interpolation of $XCH_4$ values, can also introduce correlations by spreading errors from one location to adjacent areas.

As discussed previously, transport models exhibit seasonal biases and inaccuracies in $CH_4$ profiles at high latitudes (Tsuruta et al., 2023). Most TROPOMI-based inversions use Eulerian transport models, but this work, alongside Thompson et al. (2025), employs an LPDM. LPDMs become linearly computationally expensive with increased satellite data due to the rapid growth in the number of particles modelled as resolution increases. In contrast, most Eulerian models use a fixed spatial emissions grid, making them more efficient for large datasets (although some Eulerian models have used nested grids (Lunt et al., 2019; Nesser et al., 2024)). However, they also must account for the chemical loss of $CH_4$ through reactions with OH, introducing uncertainties due to the spatial and temporal variability of different OH fields (Zhao et al., 2020). Improving the computational cost of producing footprint sensitivities is the subject of ongoing work (e.g., Fillola et al., 2023), which could result in the use of TROPOMI data on larger spatial domains – such as full-Arctic model simulations – being feasible using LPDM backward-running simulations.



## 5 Potential improvements to inversions using TROPOMI

Given the challenges of installing additional in situ measurement stations at high latitudes in the near term, improving high-latitude inversions using TROPOMI data is crucial. This section discusses various strategies to refine TROPOMI inversions in this study, and more generally in terms of model development, especially related to vertical profiles and uncertainty estimates, and improvement to the retrieval products.

Improvements could be made to the modelling of atmospheric transport by the NAME LPDM (described in Sect. 2.4).
Currently, our configuration of NAME does not simulate the atmosphere above 20 km. In our setup, particles are released up to the 20 km limit, and for levels above this, we assume the modelled $XCH_4$ concentrations match the corresponding levels in the $CH_4$ a priori profile (taken from the satellite retrieval). The TROPOMI-WFMD product is defined on 20 levels, using equal pressure intervals. The TROPOMI-OPER product, however, is defined on 12 equal pressure levels. Because of this configuration, the NAME model for TROPOMI-OPER was cut off after the 11th level at an equivalent altitude of 16 km. In
contrast, the configuration for TROPOMI-WFMD allowed full modelling up to 20 km. Consequently, a larger portion of the TROPOMI-OPER column is approximated by the $CH_4$ a priori profile. Although stratospheric contributions to the estimation of $XCH_4$ are minor compared to the tropospheric contributions (Lindqvist et al., 2024; Tsuruta et al., 2025), it can play a major role in specific conditions such as the polar vortex, and may lead to systematic biases of estimated $XCH_4$ values. An improvement for estimating TROPOMI-OPER $XCH_4$ values would be to interpolate the retrieval onto levels extending up to
20 km, similar to those used by TROPOMI-WFMD. Or, a further improvement, which would mitigate the need to replace the top levels of the modelled $XCH_4$ with the $CH_4$ a priori profile, would be to run NAME to a much higher altitude. Running NAME to a higher altitude would require a significant computational burden that was not feasible for this study, but should be considered for future work.

In this work, a priori boundary conditions are derived from CAMS v19r1 (Inness et al., 2019), which only runs until 2019.
Therefore, we apply the 2019 boundary conditions also for 2020. We acknowledge this as limitation, given the significant increase in global background concentrations from 2019 to 2020. However, the boundary conditions are optimised each month throughout the inversion, and the primary aim of this study is a comparison of satellite and surface observations in the same inversion framework, rather than to quantify absolute emissions. Future inversions will use more up-to-date versions of CAMS.

Not accounting for correlated errors could lead to an underestimation of TROPOMI model-measurement uncertainties, as
discussed in Sect. 4.5, giving the inversion overestimated confidence in the transport model and TROPOMI observations. However, implementing spatial correlations requires introducing non-diagonal matrices. As our inversion method requires the inversion of matrices with every iteration, including spatial correlations would require further computational approximations. In the near term, implementing a minimum model-measurement error, such as in Tsuruta et al. (2023), could be a practical modification to ensure that uncertainties are not underestimated. Additionally, the measurement errors provided from the
TROPOMI-OPER product are likely underestimated (mean of 3 ppb). This issue could be addressed to improve the reliability of the data product.



Enhanced filtering of TROPOMI measurements could be used to reduce noise in the retrievals. A blended albedo filter, as suggested by Wunch et al. (2011), which combines surface albedo in the NIR and SWIR to filter out snow-covered scenes, could be implemented to refine the TROPOMI data quality. Lorente et al. (2021) recommends that the albedo is limited to
0.85 to eliminate scenes that contribute to seasonal biases. This would require careful optimisation for the NSA, where there could be a large loss of data for the early and late seasons, which is a crucial for understanding NSA budget. Efforts should be made to improve retrieval quality outside of growing seasons in northern high latitudes. Since the inversions in this work were carried out, new versions of both TROPOMI-OPER and TROPOMI-WFMD have been released, which should be used in future inversions.

## 655  6  Conclusions

This study assessed the robustness of TROPOMI-derived $CH_4$ emissions using inverse modelling over the high Arctic. The focus was on tundra emissions from the North Slope of Alaska, comparing emissions derived using the TROPOMI-OPER and TROPOMI-WFMD products with those derived from in situ surface measurements, as well as multiple sensitivity tests. A comparison of TROPOMI-based inversions with surface-based inversions provides a valuable consistency check in regions lacking
ground-based column $XCH_4$ validation. However, as demonstrated in this study, both satellite and surface-based inversions are sensitive to changes in inversion inputs, which has implications for other Arctic regions without ground-based validation data, such as in Siberia.

The TROPOMI-OPER-derived tundra emissions aligned with surface measurement-derived emissions for 15 of the 19 months in the inversion period (79%). TROPOMI-WFMD-derived emissions aligned with surface-derived emissions for only 8
of 22 months (36%). Both TROPOMI products derived anomalously low emissions in August 2018, compared to those derived using surface measurements, likely due to low data density due to high cloud coverage. Future satellite missions with active LIDAR sensors, such as MERLIN, may offer higher data coverage in broken-cloud conditions due to a small Lidar ground spot (Ehret et al., 2017). Sensitivity tests revealed that TROPOMI-derived tundra emissions are much more sensitive to the choice of prior emissions compared to the surface-derived emissions, indicating a lack of robustness in our TROPOMI inversions.
Combining TROPOMI data with in situ observations increased alignment of tundra emission estimates for TROPOMI-WFMD but had minimal impact on TROPOMI-OPER.

Emissions from the anthropogenic sector were also assessed, with the largest emissions coming from fugitive emissions from the Prudhoe Bay oil fields. In this case, TROPOMI-WFMD-derived emissions agreed more closely with BRW-derived emissions than the TROPOMI-OPER-derived emissions. However, emissions derived from the combined inversions indicate
that TROPOMI observations provide more constraint to anthropogenic emissions than the BRW observations. The intra-annual variability of anthropogenic emissions in the NSA remains uncertain, as the posterior emissions from each inversion diverge from one another and from the seasonally constant prior emissions inventory.

Compared to the strong influence of prior emissions, other sensitivity tests affected emissions in fewer months and to a lesser extent, but still highlighted important sensitivities in the model setup. Changing the grid cell aggregation affected both the total



and spatial distribution of emissions in a some months, showing that grid configuration can influence both emission magnitudes and patterns. Resampling TROPOMI observations to include those only over Alaska led to higher summer emissions and indicated less constraint on the boundary conditions compared to using observations across the whole inversion domain. Finally, allowing for vertical variability in boundary condition optimisation had minimal impact on the resulting emissions.

Our work supports the recommendations of Tsuruta et al. (2023). Furthermore, we suggest the use of a minimum mea-

surement error for the TROPOMI-OPER product, given the low measurement errors provided. Additionally, we recommend that TROPOMI inversions in high northern latitudes using various inversion frameworks and transport models should be intercompared. These intercomparisons should employ the same prior emissions maps to isolate biases in the models and inversion frameworks. We encourage rigorous sensitivity testing with TROPOMI inversions, in order to evaluate the robustness of derived emissions, particularly regarding alterations to prior emissions, which, as demonstrated in this study, can have a significant im-

pact. We encourage the continued use of LPDMs in TROPOMI inversions as this becomes more computationally feasible with the implementation of emulator models. Alongside this, it is crucial to continue and expand ground-based column measurements from instruments such as AirCore or TCCON, which are instrumental in validating the vertical profiles of the satellite retrievals and transport models.

The North Slope of Alaska is a key northern high-latitude region undergoing rapid environmental change, with methane

emissions from natural sources already potentially increasing (Sweeney et al., 2016; Ward et al., 2024). While TROPOMI $XCH_4$ data is valuable for monitoring $CH_4$ emissions with inversions, the interpretation of posterior emissions requires careful consideration of the limitations to ensure accurate assessments in northern high latitudes.

*Code and data availability.* In-situ $CH_4$ mole fractions at BRW, TIK, INU and ESP are available for download from the ObsPack Data Portal at https://gml.noaa.gov/ccgg/obspack/ (last access: October 2024). The University of Bremen's WFM-DOAS v1.8 TROPOMI product is

available for download from https://www.iup.uni-bremen.de/carbon_ghg/products/tropomi_wfmd/index_v18.php (last access: July 2023). The SRON Operational v2.04.00 TROPOMI product is available for download from https://ftp.sron.nl/open-access-data-2/TROPOMI/ tropomi/ch4/19_446/ (last access: August 2023). Emissions estimates from all inversions and sensitivity tests are available at https://doi. org/10.5281/zenodo.16630729 (Ward, R. H., 2025). All the model inputs and code will be provided on request from the corresponding author.



*Author contributions.* RHW and ALG designed the research and RHW carried out the development of TROPOMI model code, experiments, visualisation, and original draft preparation with supervision from ALG. LMW contributed supervisory input throughout and provided critical review and feedback. RLT developed and supported the TROPOMI model code. EMF contributed the concept and implementation of the TROPOMI resampling routine. AT and TA provided critical review and feedback on the analysis and manuscript. All authors contributed to the manuscript through review and editing.

*Competing interests.* The authors declare that they have no conflict of interest.

*Acknowledgements.* RHW was supported by a NERC GW4+ Doctoral Training Partnership studentship from the Natural Environment Research Council [NE/S007504/1]. We thank the teams responsible for the TROPOMI instrument, which was developed through a collaboration between Airbus Netherlands, KNMI, SRON, and TNO, and was commissioned by the Netherlands Space Office (NSO) and the European Space Agency (ESA). We thank those responsible for the development of the retrieval products - the operational product by SRON and the
WFM-DOAS algorithm by the University of Bremen. We also thank Oliver Schneising and Michael Buchwitz of the WFM-DOAS team for discussions on TROPOMI data availability over the study region. We thank those at the National Oceanic and Atmospheric Administration (NOAA), Environment and Climate Change Canada (ECCC) and Finnish Meteorological Institute (FMI) that perform high-quality, continuous $CH_4$ measurements at the in-situ sites used in this study and make them publicly available to use via the ObsPack. We thank all those that have contributed to the Bristol Atmospheric Chemistry Research Group's code repository, past and present. The inversions
and analysis in this work was carried out using the computational facilities of the Advanced Computing Research Centre at the University of Bristol (http://www.bris.ac.uk/acrc/). NAME model usage was made possible by the University of Bristol's NAME license from the UK Met Office and was run on JASMIN, the UK's collaborative data analysis environment (https://www.jasmin.ac.uk). We thank the European Space Agency ESRIN Contract No: 4000137895/22/I-AG MethaneCAMP, AO/1-10901/21/I-DT AMPAC-Net, ESA AO/1-11844/23/I-NS ESA SMART-CH4, Research Council Finland projects 337552 (ACCC), 359196 (FAME) and 364975 (CHARM), and EU-Horizon IM4CA
(101183460), for financial support.



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
