# Peer review of "Uncertainty and retrieval sensitivity in TROPOMI-based methane inversions over the North Slope of Alaska"

_EGUsphere, 2025_

## Author Response (AR1)

*Below is our response to reviewer 1. Reviewer comments are in black font and our comments are in blue italics.*

1) General comments.

Methane is second most important greenhouse gas in Earth's atmosphere, and the Arctic region is a potential large source of CH4 emission. Emission detection and monitoring is therefore an interesting and relevant research question. The authors compare emission estimates based on surface-only and TROPOMI based inversions using two different retrieval products. A sensitivity study is conducted to asses the robustness of the results. Since the article adresses a relevant sicientific topic from an inversion modelling perspective, it should be published after consideration of the specific comments given below.

*Thank you for your positive review of our work, which has helped to improve the manuscript. We respond to your specific comments below.*

2) Specific comments

*) page 4, line 99: the inversion method used in this study is the same as in Ward (2024). A few lines about the general setup of this method would make the rest of the section easier to follow for those readers who are not familiar with this method and don't have the time to read the referenced paper (e.g. what is the "tuning phase" (line 115) or "MCMC sample" (line 119), or how is the uncertainty interval (line 118) calculated?).

*We substantially revised this section to provide a self-contained description of the HBMCMC inversion method, so that readers do not need to consult Ward et al. (2024). This includes the addition of an equation for the hierarchical Bayesian framework, a clearer description of the tuning phase, and an expanded explanation of the MCMC sampling procedure and uncertainty estimation. All changes are documented in the tracked changes version of the revised manuscript.*

*) page 5, line 108: in some models negative emissions are interpreted as uptake instead. Why is that not considered here?

*We use truncated normal prior PDFs for both emissions and boundary conditions to prevent the sampler from exploring non-physical or extremely large negative flux values. This choice reflects that CH4 net uptake is expected to be very small relative to emissions at the spatial and temporal scales of our inversion. In practice, uptake can still be expressed as a reduction from the prior to the posterior (i.e., posterior scaling factors below one), even though absolute negative fluxes are not allowed. We agree that prohibiting grid-cell–level negative fluxes is a limitation, and we have now clarified this point in the manuscript: "To avoid sampling large, non-physical negative emissions, we use a truncated normal prior PDF for both the emissions and boundary conditions. This constraint means that net uptake cannot be*

*represented at the grid-cell level, although reductions from the prior can still capture relative decreases in emissions."*

*) page 6, lines 156-158: I interpret regridding as calculating the average of all observations falling into the gridcell of the mentioned dimensions. This already reduces the data density of the observations, so why is additional reduction step necessary? Would using larger regridded "observations" without the additional reduction step have the same effect as the current approach?

*Regridding to the NAME resolution (0.234° lat × 0.352° lon) reduces data density but unfortunately still leaves too many observations to run NAME within a feasible time frame. The additional reduction step was necessary to reduce the number of backward simulations further, whilst still allowing us to keep the most important observations over the North Slope of Alaska. We note also that Thompson et al. (2025) faced the same issue using a different Lagrangian particle dispersion model, FLEXPART, with TROPOMI and combined spatial averaging with an error-based thinning routine to manage the computational expense.*

*) page 9 line 203-205: the $XCH4^{model}_{pert}$ is apparently valid from surface to maxlev (right-hand side of supplement eq 4 = left-hand side of supplement eq. 7). That should be made clear in the text as well, since now it's just "we have equation 2 and subtract two other terms". So these lines could be clarified by first stating why you would need $XCH4^{model}_{pert}$, and then how you calculate it.

*This is a good point and an explanation of explicitly why we need $XCH4^{model}_{pert}$ will help readers. We now include at the beginning of this paragraph: "As the inversion requires modelled mole fractions that are only dependent on NAME sensitivities, we construct a perturbed modelled column mole fraction, $XCH_{4,pert}^{model} |_t$. This represents the averaging-kernel-weighted NAME contribution over the column, from the surface to the maximum level. We obtain $XCH_{4,pert}^{model} |_t$ by subtracting all the retrieval-provided prior contributions, as follows, $CH^{prior}_{4,i}$, $A_i$ and $p_i$ are provided from the retrieval product, so the second term of Eq. \ref{eq:modelled_ch4_conv} can be subtracted from $XCH^{model}_4 |_t$. Additionally, because NAME is only run up to 20km altitude, we assume the prior profile above this level and we can also subtract this known contribution from $XCH^{model}_4 |_t$."*

*) page 9, equation 4: if the first summation is from "1 till maxlev", shouldn't the second one be from "maxlev+1 till n"? In other words, which of the two summations includes maxlev? Related to that is equation 3 and the rest of the derivation in the Supplement.

*Great point. We have updated the derivation to make it clear when the summations include maxlev and when they start from maxlev+1.*

*) page 9, line 220: please clarify the CAMS dataset that you're using and why you use an outdated version (based on the "data availability" section, you could have downloaded one in 2023). The problem with a "CAMS dataset" is that there are multiple datasets providing methane estimates, see

https://ads.atmosphere.copernicus.eu/datasets?q=methane&limit=30

Neither of the datasets listed here is from 2019. The only dataset that I found using a "v19r1" or similar versioning scheme is called "CAMS global inversion-optimised greenhouse gas fluxes and concentrations" for which the "Evaluation and quality assurance (EQA) reports" on the documentation tab point to a report by Segers, not to Inness.

*The one you found is indeed the version and type of CAMS product we used. Thank you for your thorough check. We have now been more specific about the version and type of dataset in the manuscript text, went back to find exactly when this data was accessed and added a more appropriate citation for the dataset.*

*"A priori vertical boundary conditions are derived from the CAMS global inversion-optimised greenhouse gas fluxes and concentrations v19r1 product (Segers and Houweling, 2020)."*

*"Segers, A. and Houweling, S.: CAMS global inversion-optimised greenhouse gas fluxes and concentrations, version v19r1, Atmosphere Data Store (ADS), https://doi.org/10.24381/ed2851d2, accessed: 2020-12-28, 2020."*

*The BRW inversions began in 2020, and we use the same boundaries for consistency across the two papers (this one and Ward et al., 2024). That said, it is of course pertinent to use up-to-date versions, and future inversions will use more recent boundary-condition datasets. In Section 5 we already state: "…the boundary conditions are optimised each month throughout the inversion, and the primary aim of this study is a comparison of satellite and surface observations in the same inversion framework, rather than to quantify absolute emissions. Future inversions will use more up-to-date versions of CAMS." Rerunning the inversions with a more recent CAMS inversion product is unfortunately not within the scope of this manuscript, owing to the lead author's move to a new position.*

\*) page 10, line 229: why use EDGAR v6? As you mention in section 4.1(line 492-494), emission totals over Alaska may vary significantly according to the version of the database being used.

*A fair point and more up to date versions of EDGAR are now used in current inversions. If the aim were to define an inventory estimate for Alaska, for example, it would be essential to use a more recent EDGAR version. Given that this work is framed as a sensitivity study, we consider the use of EDGAR v6 acceptable. Finally, as with the previous comment, rerunning the inversions with a more recent EDGAR version is not within the scope of this manuscript, owing to the lead author starting a new position.*

3) Technical corrections

\*) page 1, line 80: Since the acronym HBMCMC is used here, I would capitalise the word "hierarchical" as well.

*Done*

\*) page 5, line 125: the 78W should be 78N?

*Corrected*

*) page 6, line 156: please clarify if the regridding resolution is latitude x longitude or the other way around? The same holds for the other spatial resolutions mentioned hereafter.

*Latitude and longitude are added here and for all other spatial resolutions given.*

*) Supplement page 1, eq. 6: the second summation term should have n instead of 1 as it's upper level

*Well spotted, thank you.*

*) page 19, line 387-388: there is no figure 9A or 9B.

*References to these figures are removed.*

*Below is our response to Nicole Montenegro. Reviewer comments are in black and our comments are in blue italics.*

Overall, the study appears to be well justified. The research gap is clearly articulated, as is the importance of investigating this topic and the substantial challenges associated with it. The use of recent observations positions the study as a meaningful and timely contribution, particularly in regions known to contain significant gaps in observational coverage and monitoring infrastructure, and which represent an important source of methane emissions.

*Thank you for your positive and constructive comments on our work, which have improved the manuscript.*

I think it is a great contribution that the study examines uncertainty through the sensitivity of inversions to various configurations. However, I believe that these results could have been given greater prominence, especially considering that one of the stated objectives of the manuscript is precisely the quantification of uncertainty. The analysis places more emphasis on RMSE reduction than on RMSE spread, which is the quantity that more directly reflects uncertainty. I expected the manuscript to be more tightly centered on uncertainty quantification, in alignment with what the title suggests.

*We appreciate this perspective and recognise that the emphasis on uncertainty in the title could reasonably lead to expectations of a more explicit quantitative treatment. While the title is intended to reflect how the study explores the causes of uncertainty and the sensitivity in a broader sense, we do consider RMSE spread as an indicator of uncertainty as you suggest. This is addressed in more detail in response to a later comment.*

It is also noteworthy that the study demonstrates that, although TROPOMI observations provide a significant advancement, important observational gaps persist. The period from October to March remains essentially unobserved, suggesting that future missions could meaningfully support Arctic monitoring. Presenting the progress that can be achieved with the current observational record—while also highlighting the remaining potential—is highly valuable.

Below, I provide my comments section by section, indicating numbering at both general and detailed levels, and referring to specific lines when applicable:

**0. Abstract:** The abstract is concise and clearly presents the study. It effectively communicates the context, objectives, and main findings.

**1. Introduction:** The introduction appropriately presents the key elements that frame the research, identifies the gaps and challenges, and convincingly justifies the importance of studying the Arctic. Given that a substantial portion of the analysis focuses on tundra regions, I suggest including a land-cover map of the study domain, or alternatively a map distinguishing tundra and anthropogenic emission sources.

*The below figure shows the percentage coverage of different land classes from the Boreal Arctic Wetland and Lakes dataset (BAWLD) (Olefeldt et al., 2021) over the North Slope of Alaska. This demonstrates substantial coverage particularly of dry tundra, tundra wetland and lakes. Additionally, in the*

[Figure]

*supplementary information we show the separate prior emissions maps (Figure S3) and the fraction of emissions in each grid cell from each of the source types (Figure S4). If it is strongly recommended, we can add the distribution figure or similar below in the supplementary, however as the manuscript already has 10 figures in the main text and 18 in the supplementary, we are hesitant to add an additional figure.*

**2.1** The work of Ward et al. (2024) is cited frequently throughout the manuscript. In several instances, the manuscript is not sufficiently self-contained, and the citation alone does not provide enough context for the reader to fully understand the methodological or conceptual point being referenced. While it is reasonable to avoid restating the entire content of Ward et al. (2024), the manuscript should nonetheless include the essential information needed for it to be understandable on its own.

*We have substantially revised the methods section so that it can be followed without needing to consult Ward et al. (2024), as per the request of both reviewers. In addition, we assessed each reference to Ward et al. (2024) and made sure to explicitly state what aspect of that work is relevant, if it was not already stated. For example - "The measurements from these sites are filtered using a local influence filter, which excludes observations whose NAME footprints indicate they are dominated by very local surface influence, following the approach described in Ward et al., (2024)."*

**Line 99:** The methods section begins by citing Ward et al. (2024), followed by an explanation of the method. It is unclear whether the description corresponds directly to Ward et al. or whether it represents an adaptation or extension. A brief summary would help clarify this, as well as an explicit statement that further methodological details can be found in Ward et al. (2024).

*As discussed above, we have made large revisions to the methods section to make it understandable without needing to consult Ward et al. (2024).*

**2.4** Figure 3, which presents the average footprints, is not referenced in the discussion. It would be valuable to connect the inversion results to the sensitivity patterns shown in this figure, particularly to assess how the inversions based on each product relate to sensitivities over tundra and anthropogenic regions. It would also be helpful to include the sensitivity of the BRW station in the figure.

*We have added the following discussion to Section 4.1 to connect the inversion results to the sensitivity patterns: "Average NAME footprint sensitivities (Figure 3A) could potentially be used as an indicator of how strongly posterior emissions are constrained, with weaker footprints expected to produce fluxes closer to the prior. However, our results show that this relationship does not hold consistently for individual years. For example, in 2019 the TROPOMI-WFMD footprints are lower than those of TROPOMI-OPER, yet the TROPOMI-WFMD tundra emissions are least constrained to the prior in the year (Figure 5). In contrast, in 2018 and 2020 the TROPOMI-WFMD tundra emissions remain closer to the prior, even though the footprint magnitudes between the two products are more similar. However, the most pronounced spatial difference in sensitivity is that TROPOMI-OPER exhibits higher footprint values over southern NSA regions, which may allow emissions from these areas to be more strongly constrained. One explanation for the difference in the footprints between the two TROPOMI products is the number of observations remaining after the retrieval and QA filtering. The footprint sensitivities alone cannot explain the inversion result, which also depends on many factors including the retrieved column values, uncertainties, potential biases in the data, and the treatment of different vertical levels in NAME (as discussed in Section 5)."*

*We re-plotted the footprint sensitivities for each year separately (as we previously only showed the 2019 average) and restricted the average to only the months where both products had data, to avoid distorting due to differing temporal coverage. These plots are shown below. In the manuscript, we instead chose to include a plot of the average of the full period (2018-2020) where both products have data, to make this figure more representative of the whole study.*

*2018:*

[Figure]

*2019:*

[Figure]

*2020:*

[Figure]

*Finally, we have added the BRW footprint from Figure S2 to Figure 3 as suggested and added a clarifying sentence in the figure caption: "Note that the scales for TROPOMI products and BRW are not the same. Direct comparisons of magnitude between a site footprint and a satellite footprint are not meaningful because the sensitivity of TROPOMI is averaged over both the vertical column and horizontally, whereas surface sites measure a localised point, which changes the sensitivity scale."*

**2.6** The domain used for the inversions—particularly in the sensitivity tests—was not entirely clear. Figure 1 defines domain, region, and measurement sites; however, since multiple configurations are tested, it is important to explicitly clarify whether the same domain is consistently applied (if that is the case). It would also be helpful to explicitly state that the analysis focuses on the NSA region, even if this seems implied.

*Firstly, we have made the caption of Figure 1 more explicit about the study focus region: "Regions, domains and sites used in this study. The location of the focus region, the NSA, as defined in this study, is shown with a green rectangle. The Arctic inversion modelling domain used in Ward et al., (2024) spans the entire area shown, compared to the smaller Alaska inversion modelling domain (yellow rectangle) used in this study."*

*We also change a potentially confusing sentence in Section 2.1 to be more explicit - "For the current study, all of the TROPOMI inversions and sensitivity tests use a smaller "Alaska" domain (36∘N to 78∘N and 112∘E to 110∘W). All surface inversions that are compared to the TROPOMI inversions also use the "Alaska" domain. Both domains are shown in Figure 1. The impact of using a different domain on the surface inversion is discussed in Sect. 3.1."*

 Furthermore, to more directly assess uncertainty, it would have been interesting to include an experiment involving, for example, 10 perturbed prior ensembles and to evaluate the resulting spread.

*We agree that assessing sensitivity to prior assumptions is an important way to evaluate uncertainty. However, the suggested approach is not directly applicable to our hierarchical Bayesian MCMC framework, in which uncertainty is quantified explicitly through the posterior probability distributions of emissions, boundary conditions, and hyperparameters sampled by MCMC. Sensitivity to prior assumptions is instead explored through the sensitivity tests presented here, which alter prior emissions and inversion configurations.*

In Figure 4b, the color scheme overlaps with that of Figure 1, which generated confusion. For instance, the Ward domain appears in orange in Figure 4a but in black in Figure 1.

*Thank you for pointing out this confusing colour matching. We have decided to change the colours on Figure 1, to make sure they don't confuse with Figure 4.*

I recommend relocating Figure 4 to the Results section, as it corresponds to the first point addressed there.

*We have double-checked and made sure it is in the results section for the next submission.*

**3.2.1** In this subsection, the inversions correspond to the NSA, but Alaska is used as the domain—is this correct?

*Yes. this is correct. We have altered the figure captions to make this explicitly clear: "Posterior tundra emissions from the NSA for each of the BRW, TROPOMI-OPER and TROPOMI-WFMD inversions over the Alaska domain."*

**3.2.2 – Figure 6** The shift between TROPOMI-OPER and BRW-based inversions is not addressed in either the results or the discussion. Although the manuscript comments on monthly matches, the phase differences are particularly striking and, in my view, merit discussion. This may even be connected to the sensitivity analysis previously conducted for TROPOMI.

*We have added the following discussion Section 4.1 to discuss the differences between TROPOMI-OPER and BRW, particularly in the summer months where the inferred emissions diverge. This has also been combined to fit with the already present discussion of anthropogenic emissions:*

*"To further study these differences between the TROPOMI-OPER and BRW, we examined the BRW mole fractions and corresponding NAME footprints where the posterior emissions diverge. In August 2019 and 2020, where TROPOMI-OPER infers low anthropogenic emissions and BRW infers higher emissions, the posterior BRW mole fractions reproduce the observations well. However, in June 2019 and 2020, when TROPOMI-OPER infers higher anthropogenic emissions, the posterior modelled observations from BRW show a poorer fit. In some cases, this can be explained by NAME footprints exhibiting limited sensitivity over land, however, there are periods where the footprint patterns do not explain the mismatch, indicating additional factors may contribute. These diagnostics suggest that TROPOMI-OPER maybe be more sensitive to anthropogenic emissions than BRW in during some summer months. Lindqvist et al., (2024) reported a small positive bias in summer and a negative bias in autumn for TROPOMI-OPER relative to TCCON at other northern high-latitude stations, which could contribute to the lower anthropogenic emissions inferred by TROPOMI-OPER in August relative to June. However, without independent validation data over the North Slope of Alaska, it is not possible to determine whether the observed differences reflect genuine emission variability or retrieval artefacts."*

**3.3 – Figure 7** The legend lists "Uniform prior" twice. I assume that the solid line corresponds to the posterior of the Uniform-prior experiment and the dashed line to the prior.

*This figure has been updated to now say 'Uniform posterior' for the solid line, the same as in Figure S9.*

**3.3.4 – Lines 387–388** No Figures 9A or 9B are present.

*References to these have been removed.*

**3.6** To address uncertainty more directly, I recommend evaluating the spread of RMSE values for each TROPOMI product—that is, assessing how wide the RMSE range is across the experiments. The discussion currently emphasizes the RMSE magnitude for individual experiments and highlights improvements, but the variability among experiments is not addressed.

*We think this is a great suggestion, and we have included the following in results, Section 3.6: "The RMSE ranges quantify the variability among the inversion experiments. For TROPOMI OPER, the prior RMSE spans 23.7 to 33.7 ppb (a 10 ppb spread), which is larger than the corresponding WFMD prior spread of 7.1 ppb (26.1 to 33.2 ppb). This indicates that OPER is more sensitive to prior assumptions and inversion set-up. After assimilation, however, both products converge to similarly narrow posterior ranges. The OPER spread contracts to 2 ppb (16.1 to 18.1 ppb), an 80 percent reduction, and WFMD to 2.6 ppb (14.5 to 17.1 ppb), a 63 percent reduction."*

*And in the discussion, Section 4.2: "The RMSE spread across the sensitivity tests shows that TROPOMI-OPER has higher prior sensitivity than TROPOMI-WFMD, but after the inversion, both products converge to similarly narrow posterior RMSE ranges, indicating that the inversion reduces experiment-to-experiment variability."*

**3.7** Tables S1 and S2 should be included in the main manuscript.

*Done.*

**Line 435:** The reference does not correspond to Figure S15; I understand that Figure S16 is intended instead.

*Good spot, thank you.*

**6. Conclusions Line 684:** The recommendations from Tsuruta should be briefly summarized; including new citations in the conclusions is not advisable. The same applies to line 695.

*We have added the specific recommendations from Tsuruta (2023) that were discussed earlier in the manuscript, while retaining the citation to clarify that these recommendations are not our original work: "Our work supports the recommendations of Tsuruta et al. (2023), including increasing the model-measurement uncertainty or accounting for model-measurement error correlation in both space and time."*

*The references on line 695 have been removed as suggested.*

**References**

Olefeldt, D., Hovemyr, M., Kuhn, M. A., Bastviken, D., Bohn, T. J., Connolly, J., Crill, P., Euskirchen, E. S., Finkelstein, S. A., Genet, H., Grosse, G., Harris, L. I., Heffernan, L., Helbig, M., Hugelius, G., Hutchins, R., Juutinen, S., Lara, M. J., Malhotra, A., Manies, K., McGuire, A. D., Natali, S. M., O'Donnell, J. A., Parmentier, F.-J. W., Räsänen, A., Schädel, C., Sonnentag, O., Strack, M., Tank, S. E., Treat, C., Varner, R. K., Virtanen, T., Warren, R. K., and Watts, J. D.: The Boreal–Arctic Wetland and Lake Dataset (BAWLD), Earth Syst. Sci. Data, 13, 5127–5149, https://doi.org/10.5194/essd-13-5127-2021, 2021.

Lindqvist, H.; Kivimäki, E.; Häkkilä, T.; Tsuruta, A.; Schneising, O.; Buchwitz, M.; Lorente, A.; Martinez Velarte, M.; Borsdorff, T.; Alberti, C.; et al. Evaluation of Sentinel-5P TROPOMI Methane Observations at Northern High Latitudes. Remote Sens., 16, 2979. https://doi.org/10.3390/rs16162979, 2024.

Thompson, R. L., Krishnankutty, N., Pisso, I., Schneider, P., Stebel, K., Sasakawa, M., Stohl, A., and Platt, S. M.: Efficient use of a Lagrangian particle dispersion model for atmospheric inversions using satellite observations of column mixing ratios, Atmos. Chem. Phys., 25, 12737–12751, https://doi.org/10.5194/acp-25-12737-2025, 2025.